



# Quantitative Sub-Ice and Marine Tracing of Antarctic Sediment Provenance (TASP v1.0)

Jim W. Marschalek[1], Edward Gasson[2], Tina van de Flierdt[1], Claus-Dieter Hillenbrand[3], Martin J. Siegert[1,2] and Liam Holder[1]

[1]Department of Earth Science and Engineering & Grantham Institute – Climate Change and the Environment, Imperial College London, Exhibition Road, London, SW7 2BP, UK; j.marschalek18@imperial.ac.uk
[2]Centre for Geography and Environmental Sciences, University of Exeter, Penryn Campus, Cornwall, TR10 9EZ, UK.
[3]British Antarctic Survey, High Cross, Madingley Road, Cambridge, CB3 0ET, UK.

*Correspondence to*: James W. Marschalek (j.marschalek18@imperial.ac.uk)

**Abstract.** Ice sheet models must be able to accurately simulate palaeo ice sheets to have confidence in their projections of future polar ice sheet mass loss and resulting global sea-level rise, particularly over longer timescales. This requires accurate reconstructions of the extent and flow patterns of palaeo ice sheets using real-world data. Such reconstructions can be achieved by tracing the detrital components of offshore sedimentary

records back to their source areas on land. For Antarctica, however, sediment provenance data and ice sheet model results have not been directly linked, despite the complimentary information each can provide on the other.

Here, we present a computational framework (Tracing Antarctic Sediment Provenance, TASP) that predicts marine geochemical sediment provenance data using the output of numerical ice sheet modelling. The ice sheet

model is used to estimate the spatial pattern of erosion rates and to trace ice flow pathways. Beyond the ice sheet margin, approximations of modern detrital particle transport mechanisms using ocean reanalysis data produce a good agreement between our predictions for the modern ice sheet/ocean system and seabed surface sediments. These results show that the algorithm could be used to predict the provenance signature of past ice sheet configurations. TASP currently predicts neodymium isotope compositions using the PSUICE3D ice sheet

model, but thanks to its design it could be adapted to predict other provenance indicators or use the outputs of other ice sheet models.





## 1. Introduction

### 1.1 Motivation and aims

Antarctic ice flow pathways and ice sheet extent in the geological past can be reconstructed using sediment

provenance studies, which trace marine detrital sediments to their constituent source rocks on the continent (e.g.
Cook et al., 2013; Licht and Hemming, 2017; Wilson et al., 2018; Marschalek et al., 2021). However,
conclusions about ice sheet extent and flow patterns drawn from such data are typically qualitative. This is
because, when viewed in isolation, sediment provenance records can only predict where subglacial erosion was
taking place, but cannot distinguish between changes in ice extent, changes to subglacial conditions which

influence erosion rates, or changes to glacimarine sediment transport processes (Wilson et al., 2018; Golledge et
al., 2021). Although such complexities are often considered, interpretations of sediment provenance records are
typically limited to a 'nearest-neighbour' type approach, where changes in provenance are attributed to a simple
advance/retreat of the ice sheet margin in the vicinity of the core site (e.g. Cook et al., 2013). Numerical
modelling offers the potential to make more quantitative estimates of the ice sheet state required to generate

observed sediment provenance signatures by predicting the expected provenance of detritus at the ice sheet
margin (Aitken and Urosevic, 2021). In this study, we develop an algorithm (Tracing Antarctic Sediment
Provenance, TASP) to predict marine geochemical sediment provenance data from the output of numerical ice
sheet modelling.

Although the overall aim of TASP is to provide more quantitative information on past ice sheet configurations

by predicting the provenance signature of these ice sheets, we here focus on application of TASP to the modern
Antarctic ice sheets (i.e., East Antarctic Ice Sheet, West Antarctic Ice Sheet (WAIS) and Antarctic Peninsula
Ice Sheet). This is because an ability to accurately represent the modern system is a pre-requisite for reconstructing
sediment provenance in the past. For the modern Antarctic ice sheets, the accuracy of TASP can be more easily
evaluated as the spatial distribution of provenance proxy measurements in marine sediments is far more

complete in recent (i.e., late Holocene to modern) surface sediments than in older, more difficult to date,
sediments (e.g. Simões Pereira et al., 2018). This is accompanied by the ability to directly measure ice surface
velocities and ice thicknesses; ice flow drainage pathways and erosion rates are therefore far more precisely
constrained than is possible for past ice sheets. Modern oceanographic data mean marine processes transporting
glaciogenic detritus can also be more accurately simulated at the present day. By predicting the provenance

signature of the modern Antarctic ice sheets and comparing this to measured seafloor surface sediments, we
demonstrate the accuracy of TASP and show that applying it to past ice sheets should produce useful results.

To trace debris transport, a sediment provenance proxy is required. We here choose detrital neodymium (Nd)
isotopes. This is a powerful provenance tracer which can be expressed as a single value, with widespread
application in Antarctic sediments (Farmer et al., 2006; van de Flierdt et al., 2007; Roy et al., 2007; Cook et al.,

2013; Pierce et al., 2017; Simões Pereira et al., 2018; Marschalek et al., 2021). However, it is important to note
that TASP could easily be adapted for any sediment provenance tracer, such as clay minerals (e.g. Ehrmann et
al., 2011), heavy minerals (e.g. Hauptvogel and Passchier, 2012), elemental concentrations (e.g. Monien et al.,
2012), clast petrography data (e.g. Sandroni and Talarico, 2011) or detrital mineral ages (e.g. Licht et al., 2014).



## 1.2 The structure of TASP

TASP estimates relative erosion rates and reconstructs the transport pathways of debris from beneath the ice sheet to the seafloor surrounding the Antarctic continent. This includes representations of the approximate trajectories of debris at the base of the ice sheet, below/in ice shelves, in icebergs/ocean surface currents, in ocean bottom currents and in gravity flows. The primary distinction between our approach and previous computational estimates of Antarctic sediment provenance (Aitken and Urosevic, 2021) is our inclusion of detritus transport in the ocean. This is a critical component controlling sediment provenance, as most sediment core records are not directly located at the ice sheet margin today and their distance from the ice sheet margin likely varied significantly in the past. Transport of glaciogenic detritus in the ocean must therefore be considered. TASP approximates methods of marine detritus transport by splitting them into three key, interacting processes: iceberg rafting/transport in surface currents (Section 2.2), bottom current transport (Section 2.3), and gravitational downslope processes (Section 2.4) (Fig. 1).

In summary, TASP is structured in a broadly similar way to how a debris particle might travel from source to sink (Fig. 1):

1.  Debris is generated by glacial erosion, with the relative amount of debris produced at any location proportional to the product of the basal shear stress and basal ice velocity. Areas with higher basal shear stresses/velocities will produce more debris. Debris is assigned a Nd isotope signature based on a geology-based map of the subglacial source rocks (Appendix 1). Sediment is transported to the grounding zone using basal ice sheet velocities and a flowline algorithm. Because of the time integration of ocean sediment cores (typically $10^3$-$10^5$ years), we ignore deposition and remobilisation along the flowline.

2.  As results are compared to marine sediment cores, debris movement from the ice sheet grounding zone to marine sediment core locations must be determined. Assumptions are therefore made regarding the distribution of debris in the ice column and the rate this debris is released from icebergs/ice shelves. These are combined with ice shelf velocities and ocean surface current trajectories to suggest how debris is initially moved from the grounding zone to marine sites.

3.  Ocean bottom current velocities are used to determine where these may redistribute debris and the likely trajectories these currents take.

4.  If a slope threshold is reached, downslope transport of debris is allowed for. This method is iterated once with the bottom current method, so debris transport can be a combination of all three processes.

5.  These different estimates are combined to make a 'best estimate' map of seafloor Nd isotope compositions for the modern Antarctic seafloor. This estimate is compared to measured seafloor surface samples around West Antarctica and parts of East Antarctica to demonstrate the accuracy of the algorithm.



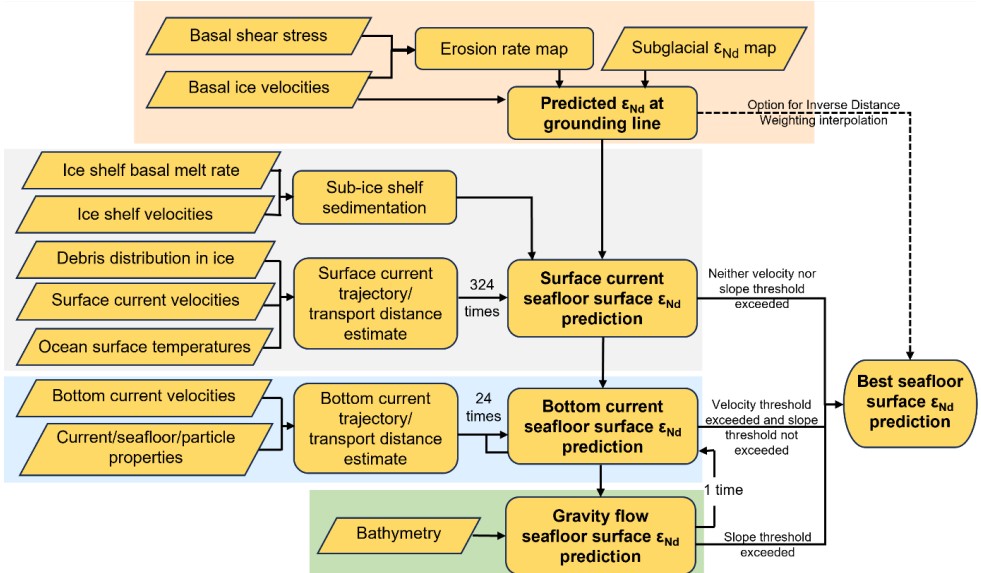

**Figure 1. Flowchart outlining the structure of TASP. Coloured boxes indicate the components contained in the 'terrestrial.m' (orange), 'surf_currents.m' (grey), 'bot_currents.m' (blue) and 'grav_flows.m' (green) subfunctions.**

Beyond this overall structure, it is important to understand at the outset that:

- TASP does not make any attempt to quantify absolute amounts of debris/sediment. This is because, for provenance purposes, relative amounts of debris are all that is required; for a given offshore sediment sample, the fraction sourced from each rock type in the catchment is what is measured.

- TASP does not step through time simulating debris transport. Samples from sediment cores typically integrate provenance signatures over long timescales, making time-evolving particle tracking at a continental scale unfeasible. Instead, TASP takes a 'snapshot' approach throughout, assuming the system is in steady state.

- TASP is written in MATLAB and is external to the ice sheet model code. This approach provides flexibility, with the opportunity to apply it to various existing model experiments. TASP is not model-specific and could be applied to any ice sheet model output, providing key variables (bed elevation, basal shear stress and basal ice velocities with horizontal directional components) are saved.

## 2. Methods

### 2.1 Neodymium isotope input data

Neodymium isotope compositions represent an excellent provenance tracer in Antarctic sediments. The decay of $^{147}Sm$ to $^{143}Nd$ relative to the stable $^{144}Nd$ abundance gives rocks a $^{143}Nd/^{144}Nd$ signature which is a function of the time since a magma was separated from the mantle and the initial composition of the resulting magmatic rock (Goldstein and Hemming, 2003). This ratio is typically reported in epsilon notation ($\varepsilon_{Nd}$); i.e., in parts per 10,000 compared to the modern Chondritic Uniform Reservoir (CHUR; Jacobsen and Wasserburg, 1980):





$$\varepsilon_{Nd} = \left( \frac{(^{143}Nd/^{144}Nd)_{sample}}{(^{143}Nd/^{144}Nd)_{CHUR}} - 1 \right) \times 10^4 \qquad (1)$$

This means rock groups of different lithologies and ages can be differentiated based on their $\varepsilon_{Nd}$ value, even if the initial magmatic rock is metamorphosed or eroded. Neodymium isotopes are a powerful sediment provenance proxy as all rock-forming minerals incorporate Nd and Sm into their structure, meaning nearly all
rock types will be accounted for and are integrated in the provenance signal. Furthermore, $\varepsilon_{Nd}$ values are generally unaffected by grain size sorting (Garçon et al., 2013) and the combined provenance signal is expressed simply as a single number. These factors make $\varepsilon_{Nd}$ values ideal for tracing in an algorithm tracing sediment provenance.

Geographically, we examine the region offshore of West Antarctica and adjacent areas of East Antarctica (from
approximately 60°W to 140°E; Fig. 2). This spans the location of many existing detrital Nd isotope provenance studies (e.g. Cook et al., 2013) and compilations of the $\varepsilon_{Nd}$ values of continental rocks (e.g. Simões Pereira et al., 2018; Marschalek et al., 2021). Although TASP estimates provenance all around Antarctica, useful results are limited to this extent of compiled literature Nd isotope compositions. Expanding the useful area of the domain would require additional compilation of bedrock literature data and assumptions regarding subglacial geology
(Appendix 1).

To assign eroded detritus an $\varepsilon_{Nd}$ value for comparison with offshore records, the first requirement is a spatially continuous estimate of $\varepsilon_{Nd}$ for the study area below grounded ice (Fig. 2a). Constructing this map required the compilation of literature $\varepsilon_{Nd}$ data for different rock types (see Appendix 1 for a detailed description). This process was aided by an existing compilation of $\varepsilon_{Nd}$ data from the Pacific sector of West Antarctica (Simões
Pereira et al., 2018, 2020), extended to the Ross Sea sector (Marschalek et al., 2021).

To assess the accuracy of our sediment provenance tracing algorithm, results were compared to $\varepsilon_{Nd}$ values measured in recent seabed surface sediments (Fig. 2a). These data were compiled from all known literature sources in our study domain (Walter et al., 2000; Roy et al., 2007; van de Flierdt et al., 2007; Pierce et al., 2011; Cook et al., 2013; Struve et al., 2017; Simões Pereira et al., 2018, 2020; Carlson et al., 2021; Shao et al., 2022).
New seafloor surface sediment $\varepsilon_{Nd}$ data from the Ross Sea were also included to improve spatial coverage in this area (Table A2).







**Figure 2. a) Map of inland ε_Nd values used as an input for the provenance tracing (Appendix 1) and interpolated ε_Nd values within 200 km of a marine surface sediment sample with a measured ε_Nd value. Seafloor surface sample**

**locations are shown as white circles and volcanoes with reported subaerial eruptions in the historic record are shown as red triangles (Patrick and Smellie, 2013). The MEaSUREs grounding line and modern ice (shelf) fronts are indicated using solid grey and black lines, respectively (Rignot et al., 2013; Mouginot et al., 2017). b) The uncertainty level of ε_Nd value estimate, displayed on BedMachine bathymetry (Morlighem et al., 2020). Uncertainty levels are classified either as: exposed rock with ε_Nd data (black); areas with subglacial geology constrained by**

**gravimetric/magnetic/geomorphological etc. data (brown); areas with few direct constraints, limited to isolated subglacial samples obtained by drilling through the ice (grey); and areas without any rock type constraints, filled using Kriging (white). Volcanic seamounts and islands (e.g. Lawver et al., 2012; Kipf et al., 2014) are circled in red. These features are displayed because they potentially supply additional radiogenic detritus to marine sediments which is unaccounted for in our algorithm. The Southern Boundary of the Antarctic Circumpolar Current (SBACC)**

**is shown as a black dashed line (Orsi et al., 1995).**

### 2.2 Subglacial debris generation and transport

The starting point of the TASP algorithm is an ice sheet model output, which is used to estimate the generation and subglacial transport of debris (Fig. 1). Here, we use a simulation of the modern ice sheet from the model PSUICE3D, which is a finite-difference numerical model with hybrid ice flow dynamics (Pollard and DeConto,

2012). The simulation used (DeConto et al., 2021) closely matches modern observations. It was run at 10 km resolution, although TASP automatically adjusts to match coarser resolutions.

Following the approach of Pollard and DeConto (2019), TASP approximates erosion rate ($E$) as proportional to the product of basal velocity ($u_b$) and basal shear stress ($\tau_b$) from the ice sheet model:

$$E = \tau_b u_b k \tag{2}$$

The 'quarrying' coefficient $k$ is a parameter dependant on properties including the erodibility of different rock types and other parameters which are unaccounted for. Here, a spatially uniform value of 5.1 x 10⁻¹⁰ was used for $k$ as suggested by Pollard and DeConto (2019). As we are concerned only with relative, and not absolute, quantities of detritus the value used will not impact results. Experiments were also performed with spatially variable $k$ values for different sectors of Antarctica tuned to the quantity of marine sediment deposited on the

Antarctic continental margin since 40 Ma (Pollard and DeConto, 2019). However, this yielded a negligible change to the match with seafloor surface sediment ε_Nd values compared to results when using a constant $k$.

TASP differs from the method used to predict sediment generation from numerical modelling by Aitken and Urosevic (2021), who used the squared basal velocity (capped at 600 m/yr) as an erosion rate proxy. Furthermore, Aitken and Urosevic (2021) used a 20 km resolution instead of our 10 km ice sheet model

resolution, as well as a Bayesian inference approach to account for uncertain geological variation.



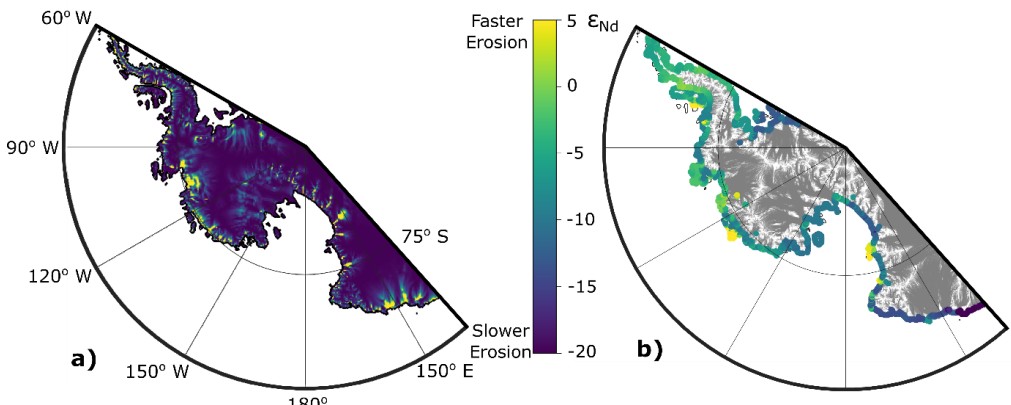

**Figure 3. a) Erosion rate map used to seed start locations. Absolute erosion rates are not given as these are uncertain and not required to predict offshore provenance. b) Grounded ice streamlines (grey), with points at the end of each streamline coloured according to the $\varepsilon_{Nd}$ value at the streamline start point. The colour bar in the centre shows both the erosion rate in panel a) and the $\varepsilon_{Nd}$ values in panel b).**

From the site of debris generation at each grid cell, sediment packages were traced to the grounding line using the MATLAB 'streamline' function and the directional components of basal ice velocity (Fig. 3b). The 'streamline' function calculates curves tangential to the velocity vector field (i.e., streamlines) using a standard Euler method from input seed locations. Seed locations were selected based on a user-defined erosion rate threshold, which is set to 0 to include all cells beneath grounded ice. Increasing this threshold would lead to faster run times, at the expense of neglecting erosion in areas with low erosion rates. The streamline function is also used in the marine components of TASP (see Sections 2.3 and 2.4) and is responsible for most of the computational demand. Using default settings, 8 CPUs and 64 GB of memory required a run time of ~6-8 hours on the Imperial College London HPC cluster. Fewer CPUs and less memory could be used, although this would increase the run time.

Once the streamlines were calculated, the $\varepsilon_{Nd}$ values at the grounding zone end locations were set using the $\varepsilon_{Nd}$ values at the streamline start locations defined by the $\varepsilon_{Nd}$ input map (Fig. 2a) and the erosion rate map (Fig. 3a). Typically, multiple streamlines arrive at a given grounding zone end cell. The final $\varepsilon_{Nd}$ values of grounding zone end cells were therefore calculated by multiplying the $\varepsilon_{Nd}$ values of each streamline start location by the erosion rate value of the start location, then summing this for all streamlines at a grounding zone end cell. This was then divided by the summed erosion rate of all the streamlines arriving at the grounding zone end cell. The $\varepsilon_{Nd}$ values at a given cell in the grounding zone thus reflect integration of the subglacial detritus at all locations upstream weighted to the relative erosion rate.

Particle transport has been included in many previous studies seeking to examine glacier flow or validate models (e.g. Clarke and Marshall, 2002; Jouvet and Funk, 2014). However, these studies typically focus on tracing ice movement within the ice column for ice core dating (e.g. Clarke and Marshall, 2002), or on alpine glaciers which have a very different spatial scale and topographic/climatological setting (e.g. Jouvet and Funk, 2014). As, in Antarctica, we are primarily concerned with debris generated at the ice sheet bed, we neglect



vertical transport. TASP also assumes all debris generated is advected to the ice sheet margin (i.e., no significant
subglacial sedimentation occurs). Furthermore, steady-state ice flow is assumed, allowing any change to ice
flow trajectories to be neglected. Given the setting and spatial scales of interest here, these approximations are
sufficient to capture the broad-scale trajectories of debris under the ice sheets.

The approach used here does not account explicitly for detritus transport in subglacial hydrological networks.
However, in Antarctica, the general lack of surface melt means only basal meltwater feeds subglacial
hydrological networks, resulting in very low sediment fluxes, even during subglacial lake drainage events
(Hodson et al., 2016; Alley et al., 2019). Furthermore, subglacial hydrological networks are unlikely to deviate
significantly from ice flow vectors at the continental scale of interest here (Willis et al., 2016). Debris transport
by subglacial hydrology is therefore neglected in this study, which focusses on the modern ice sheet. However,
we note that if significant surface melt occurred during warmer periods in the geological record, then subglacial
hydrological networks may have been more important for transporting detritus. Reconstructing such debris
transport is, however, beyond the current scope of TASP.

There are continuing difficulties in quantifying processes of erosion and entrainment of subglacial debris,
despite extensive study of these topics (Alley et al., 2019). We acknowledge that more sophisticated methods
have been previously applied to model erosion beneath alpine glaciers (e.g. Ugelvig et al., 2018; Magrani et al.,
2022). However, it is not feasible to apply the complexity of these erosion models to the much larger Antarctic
ice sheets given the large variations in surface slope, substrate properties and thermal regime, as well as the
much larger scale.

**2.3 Ocean surface currents and iceberg/ice-shelf rafting**

**2.3.1 Debris distribution in the ice column**

Before predicting how debris is deposited in the marine realm, it is important to consider the distribution of the
debris concentration in the ice column as this is a primary control on marine debris transport. Observations fully
penetrating the ice column are rare but show that, at least outside of mountainous regions, most glaciogenic
debris is concentrated in the basal layers of the ice column, typically the lowermost 2-15 metres (Gow et al.,
1979; Christoffersen et al., 2010; Shaw et al., 2011; Pettit et al., 2014). Empirical relationships of the
distribution of debris within this basal ice layer have been formulated, showing – from the base upwards - a
constant debris content for several meters followed by a sharp exponential decay (Yevteyev, 1959), but very few
data are available to suggest how typical this relationship is around Antarctica (Drewry and Cooper, 1981).
Indeed, it can be assumed to vary significantly between localities depending on factors such as how the debris is
incorporated (i.e., shearing and ice deformation vs. basal freeze on; Drewry and Cooper, 1981). Ice sheet basal
conditions, including melting/freezing rates and the vertical component of ice flow, thus govern the thickness
and nature of this debris-rich basal ice layer (Licht and Hemming, 2017). Conditions at the ice sheet bed are
therefore critical (Dowdeswell and Murray, 1990), and result in substantial variations in the distribution of
debris in basal ice around Antarctica before the ice reaches the grounding zone.

Above this debris-rich basal ice layer, the debris concentration is strongly influenced by the glaciological
setting. For instance, due to a relatively high number of rock outcrops, smaller icebergs calved from glaciers
draining through the Transantarctic Mountains or mountainous parts of the Antarctic Peninsula carry more

supraglacial debris and englacial debris higher in the ice column than larger tabular icebergs calved from the Ross or Filchner-Ronne ice shelves, which may predominantly carry basal debris (e.g., Anderson, 1999). These significant and poorly understood complexities mean estimating a variable debris distribution down the ice

column for different regions of Antarctica is beyond the scope of this study. Debris distribution in TASP is simplified by assuming it is the same in all icebergs around the continent, although we acknowledge that accounting for a variable debris load would improve the accuracy of the results.

To reflect the potential for limited concentrations of debris higher in the ice column, we include a very low minimum englacial debris content. This is included because (i) supraglacial debris is very rare, but occasionally

present in Antarctica (e.g., Evans and Ó Cofaigh, 2003); (ii) englacial debris above basal debris-rich layers has been observed in multiple locations (e.g. Nicholls et al., 2012; Winters et al. 2019; Smith et al., 2019); and (iii) iceberg rafted debris (IBRD) can be transported for hundreds to thousands of kilometres offshore (Dowdeswell and Murray, 1990; Dowdeswell et al., 1995; Gil et al., 2009). This background debris concentration is set to $10^{-5}$ times the maximum debris load at the base of the ice column based on an assumption of a ~10% debris

concentration in basal ice vs. ~0.001% higher in the ice column (Dowdeswell and Murray, 1990). We find TASP is insensitive to the precise quantity of debris higher the ice column used, but neglecting this debris entirely substantially worsens the match with seafloor surface sediment measurement (Fig. 4).

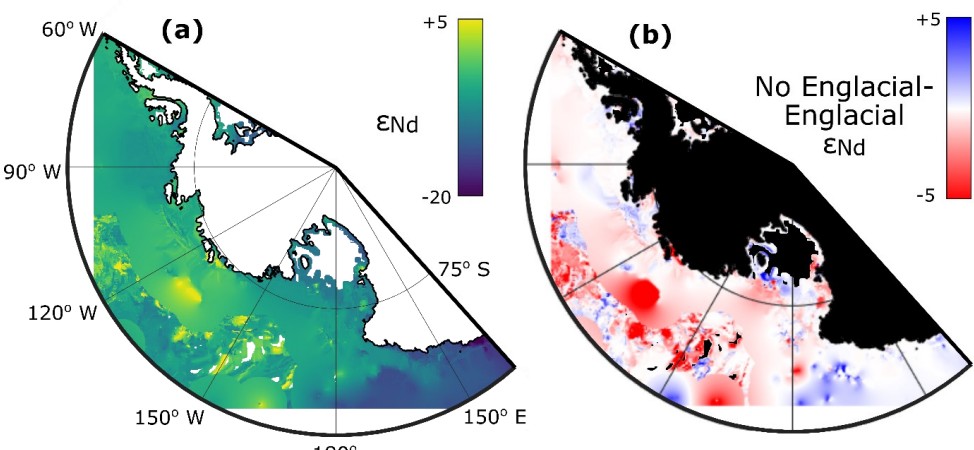

**Figure 4. Impact of assuming all debris is contained in a basal layer. a) The result assuming there is no englacial**

**debris. White areas offshore lack data. b) Difference between the best match estimate $\varepsilon_{Nd}$ values assuming englacial debris is absent (panel a) or present (Fig. 10g). Black offshore areas lack data.**

To summarise, TASP assumes debris in the ice column at the grounding zone is distributed predominantly in a debris-rich basal layer, above which there is a very small, but finite debris concentration. From an initial debris content at the ice-bed interface ($D_0$), the debris concentration in the ice ($D$) at a given height above the ice base

($Z$) is assumed to decay linearly up the ice column, reaching a minimum 'englacial' concentration ($D_e$):

$$D = \ \max\left(D_e, D_0\left(1 - \frac{Z}{Z_0}\right)\right) \tag{3}$$



Where $Z_0$ is the debris-rich basal ice thickness in metres. A value of 4 m was selected for $Z_0$ based on tuning within the approximate range of observed thicknesses (Section 2.7.2). TASP was found to be relatively insensitive to the value chosen. As summed erosion rates from the terrestrial component of TASP are saved at

the end cells of ice flow trajectories, $D_0$ is set based on the amount of eroded material reaching these grounding zone cells. $Z$ is a cumulative product of melt rate and the amount of time elapsed, as described in Section 2.3.4.

### 2.3.2 Representation of ice shelves

Once in contact with the ocean, ice may begin to melt, releasing debris from the ice column. In many locations around Antarctica, melting of basal ice may begin before icebergs calve due to the presence of floating ice

shelves. Melting at the grounding zone can lead to high accumulation of previously frozen-on basal debris or deforming subglacial till in sedimentary landforms, such as a till delta or grounding-zone wedge (e.g., Alley et al., 1989; Batchelor and Dowdeswell, 2015). Important influences on sub-ice shelf sedimentation include the seaward extent and morphology of the ice shelf, the presence of pinning points which can allow freeze on of new sediment, and the temperature of ocean water reaching the base of the floating ice which governs basal

melt/freezing rates (e.g., Smith et al., 2019). These processes often lead to basal detritus being deposited, although debris near the base can also rise in the ice column if net freezing and/or surface ablation occur (Kellogg & Kellogg, 1988; Kellogg et al., 1990; Nicholls et al., 2012).

Beneath ice shelves, debris frozen into the ice will continue to follow ice-flow trajectories, so are set in TASP by the ice sheet model. Ocean currents beneath ice shelves are very difficult to observe, thus poorly constrained.

TASP therefore assumes that there is no debris transport by ocean currents beneath ice shelves, with debris instead following modelled ice flow trajectories and melting out (or refreezing debris-free ice) at a rate controlled by the estimated sub-ice shelf melt rate over the 2010-2018 period (Adusumilli et al., 2020). In the code, ice flow velocities are effectively treated as very slow ocean current velocities, allowing a realistic pattern of melting (thus sedimentation) and freezing beneath ice shelves.

### 2.3.3 Surface current trajectories

Once calved, iceberg paths are a function of surface ocean current velocity, wind velocity and iceberg size, but surface current velocities dominate, particularly for large icebergs (Rackow et al., 2017; Wagner et al., 2017). The ORAS5 ocean velocity reanalysis dataset (Zuo et al., 2019; Copernicus Climate Data Store, 2021) was therefore used to approximate the pathways that icebergs would be expected to take after calving. This velocity

field was used as an input for the MATLAB streamline function, with the end cells of the grounded ice sheet streamlines (i.e., the grounding zone) used as seed locations for the ice shelf/ocean streamlines.

Whilst it is reasonable to assume the velocities of debris beneath an ice shelf are constant through time, using a single snapshot of ocean velocities is not accurate. This is because sea ice variability and weather patterns create substantial seasonal and interannual variations in iceberg pathways estimated from ocean surface current

velocities. To account for this effect, debris trajectories were calculated 324 times: once for each monthly mean velocity in the ocean reanalysis data spanning January 1993 to December 2019 (27 years). Debris transport was assumed to follow the trajectories produced by each velocity field. Experiments on the number of months required showed using additional years improved the match between seafloor surface sediment data and TASP's



predictions, plateauing beyond ~20 years. 27 years is therefore considered ample to accurately capture most
annual/interannual variation in iceberg trajectories.

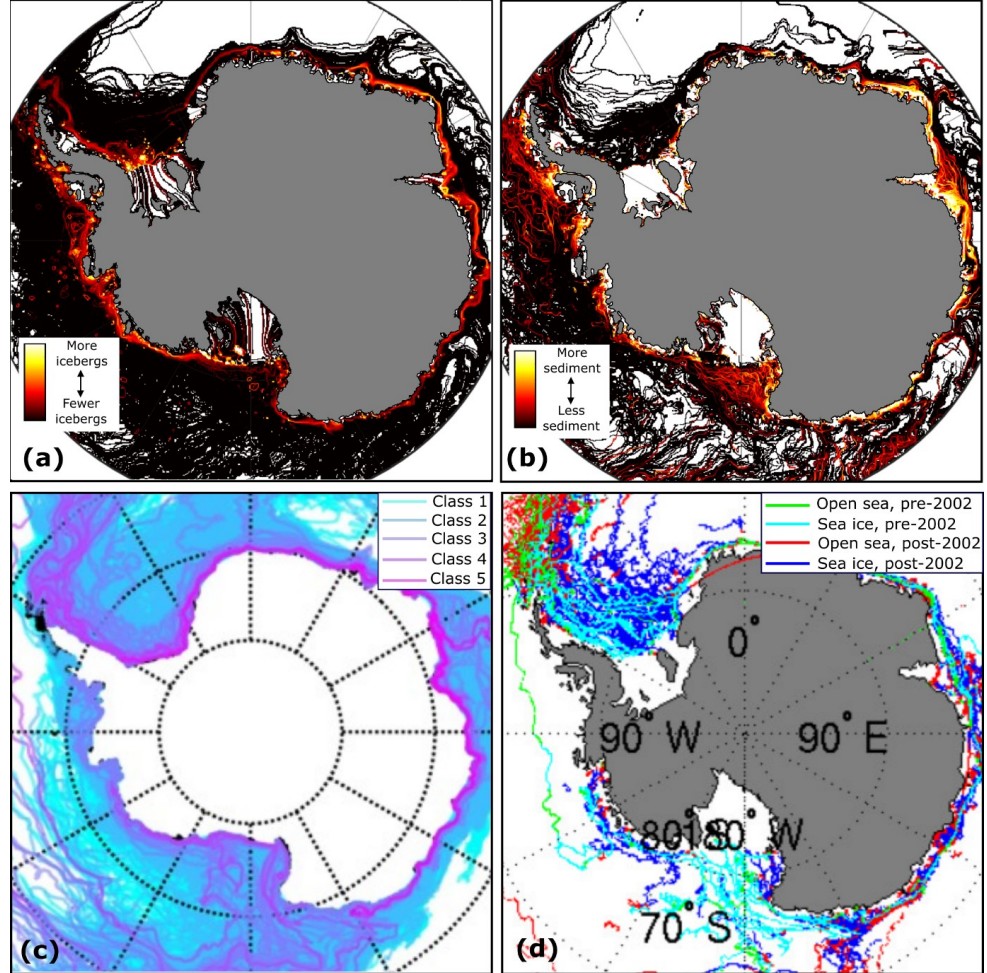

**Figure 5. a) Heatmap of ocean surface velocity paths, which are assumed to approximate iceberg paths. 'Hotter'
colours highlight areas where icebergs pass through more frequently. b) The relative quantity of iceberg-rafted
sediment deposited by our representation of icebergs (unitless). c) Modelled iceberg trajectories accounting for**
**additional influences such as wind velocities and sea ice (Rackow et al., 2017). Colours represent different iceberg size
classes (1 = 0-1 km², 2 = 1-10 km², 3 = 10-100 km², 4 = 100-1000 km² and 5 = 1000-5000 km²). d) Observed trajectories
of large (>5 km across) icebergs from the U.S. National Ice Centre and Brigham Young University (Stuart and Long,
2011), presented in Tournadre et al. (2016). The broad pattern of iceberg movement produced by our algorithm in (a)
agrees well with modelled (c) and observed (d) iceberg trajectories in our study area.**

We verify that our iceberg pathways are traced correctly by comparison to observations of iceberg movement.
Our results are broadly similar to observed iceberg tracks and studies modelling iceberg drift and distribution


(Fig. 5). In addition, the spatial distribution of smaller icebergs observed from ships matches with the distribution we produce (Orheim et al., 2021).

We note that the ORAS5 ocean velocity reanalysis dataset only extends to 80°S, which does not quite reach the coast along parts the Ross and Weddell seas. This produced a gap in the velocity field. Simply interpolating available surface ocean velocities did not produce plausible flow directions near the ice sheet margin in the regions not covered by the reanalysis product. Consequently, the mean of the interpolated ocean velocity data and extrapolated ice velocities beyond the ice sheet margin was used, after scaling the latter to the mean of local ocean velocities so they are a consistent order of magnitude.

Trajectories are not consistent from ~90°E westward to ~60°W, because the path of the Antarctic Coastal Current strays outside of our model domain (and study area) at ~90°E (Fig. 5; model domain shown in Fig. 2). A larger domain would therefore be required to study areas between 90°E and 60°W (e.g. Iceberg Alley in the Weddell-Scotia confluence).

### 2.3.4 Debris distribution along surface current trajectories

Once the trajectories of surface currents have been determined, the debris distribution along each one can be predicted. This is important because where multiple trajectories cross the same ocean cell, there must be a realistic weighting to the different source locations. To achieve this, the subglacial erosion rate values from the terrestrial component feeding into each cell at the grounding zone are summed and used to weight the initial debris concentration in each ocean surface current starting cell.

In the ocean, modern melt rates in TASP are calculated using sea surface temperatures ($T_s$) corresponding with the monthly ocean surface velocity data used (Zuo et al., 2019; Copernicus Climate Data Store, 2021). Melt rate ($M_b$) will vary as a function of ocean temperature; we parameterise this from the empirical equation of Russell-Head (1980):

$$M_b = 2.08 \times 10^{-7}(T_s + 1.8)^{1.5} \tag{4}$$

$M_b$ is calculated for every ocean cell for each month in the ocean dataset. TASP then loops over each cell in each surface current trajectory, calculating $Z_{(n)}$ (the thickness of basal ice melted in cell $n$) as:

$$Z_{(n)} = Z_{(n-1)} + \left(\frac{M_{b(n)} d}{v_{mag(n)}}\right) \tag{5}$$

Where $Z_{(n-1)}$ is the thickness of ice already melted from the 'iceberg' in the previous cell, $d$ is the distance travelled from the last cell, and $v_{mag(n)}$ is the magnitude of the velocity in the given cell. The debris yield for the given cell is then given by Equation 2.

In reality, the relative amount of debris release over time from individual icebergs will depend on iceberg size and frequency of overturning, and as such may be episodic (Drewry and Cooper, 1981; Dowdeswell, 1987; Dowdeswell and Murray, 1990). However, we verify that our equations accurately represent iceberg melt by inputting typical Southern Ocean values for $T_s$; this produces debris yields over time comparable to those of studies with a more rigorous treatment of iceberg movement and melt (Dowdeswell and Murray, 1990; Hopwood et al., 2019).





To convert this relative debris amount into an estimate of the Nd isotope composition for each ocean cell, the Nd isotope composition of glaciogenic debris at each ice sheet grounding zone cell was projected along the iceberg paths and weighted based on the debris content of the grounding zone cell and the amount of basal ice melted (Equations 2&5). Each ocean cell is therefore assigned two values for each streamline: a) an $\varepsilon_{Nd}$ value, and b) a relative debris amount. For each month in the surface ocean velocity/temperature dataset, a map of Nd isotope compositions is then calculated based on the debris amount saved for each streamline and the $\varepsilon_{Nd}$ values. A mean is then taken of the 324 monthly estimates to produce the final 'surface current' Nd isotope composition estimate.

Areas with no iceberg tracks over them, but within 40 km of an iceberg track, were filled using interpolation. This interpolation somewhat accounts for the potential for variations in ocean surface currents which are unresolved here. Furthermore, it is likely that over the centuries integrated within most sediment samples of centimetre-scale thickness, there would be transport of icebergs over more of the seafloor than captured in the 27 years of data used here.

A given 'iceberg' is not, therefore, modelled in a true time-evolving manner as its trajectory is calculated based on a single month's surface current velocity data. Although unrealistic, this simplification was necessary to avoid detailed modelling of icebergs which would need to include additional factors, such as morphology, wind stress etc. (e.g. Rackow et al., 2017), which are beyond the scope of this study. Despite this simplification, TASP captures the expected patterns in surface current movement offshore (Fig. 5). Debris from icebergs more proximal to their source location will dominate the $\varepsilon_{Nd}$ signature, but where more local icebergs are absent, there is the potential for far-travelled icebergs to influence $\varepsilon_{Nd}$ values of the seafloor sediments.

Sea ice conditions impact iceberg movement and melt, leading to seasonally changing detritus transport in surface currents. We do not explicitly account for sea ice movement but note that the far lower sea surface temperatures in the winter months, when icebergs usually calve less frequently and are frozen into sea ice, will weight our results near the coast towards the summer months.

### 2.3.5 Suspended sediment

Until now, it has been assumed that debris is transported entirely within ice, which is not realistic as fine-grained detritus can also derive from meltwater plumes rising from beneath the ice shelf base at the grounding zone (e.g. Lepp et al., 2022). Meltwater plumes can deposit sediment tens of kilometres away from the grounding zone and will track ocean currents (e.g. Smith et al., 2019, and references therein). Furthermore, once melted out of the ice shelf/icebergs, very fine grained IBRD will not instantaneously fall out of the water column. Instead, silt particles (on the order of 5-50 μm) with slower settling velocities will be transported horizontally as they sink down through the water column over periods of days to months, leading to potential transport distances of tens to thousands of kilometres, depending on grain size and current speed (Azetsu-Scott and Syvitski, 1999).

Meltwater plumes (and fine-grained debris melted out of icebergs) can travel at varying heights in the water column, depending on the relative densities of the plume and surrounding ocean water. However, we argue these debris transport mechanisms will track ocean flow patterns and hence are somewhat represented in TASP by the





use of both surface and bottom current approximations (see also Section 2.4). Given that our $\varepsilon_{Nd}$ value estimate

using ocean surface currents represents not only IBRD, but also potential settling of fined grained detritus derived from icebergs or meltwater plumes, we refer to the estimate derived from tracking surface currents as the 'surface current' method rather than the IBRD method.

### 2.4 Bottom currents

Iceberg rafting and surface currents are not responsible for all transport of glacimarine detritus. One key process

operating, especially on the continental rise, is the transport and deposition of predominantly silt- and clay-sized detritus by bottom currents. These currents typically flow parallel to the continental slope where they can influence the movement of detrital particles already in suspension. Bottom currents do not only deposit sediments as "contourites" but can also erode and redistribute particles from the seabed surface if their velocities are sufficiently high. As well as in the deep ocean, bottom current velocities can occasionally be sufficiently

high to remobilise sediment on the continental shelf (Ha et al., 2014; Jenkins et al., 2018).

In TASP, bottom currents were defined as the deepest velocity in the ocean reanalysis product (Zuo et al., 2019; Copernicus Climate Data Store, 2021). Due to very sparse in-situ measurements, knowledge of bottom current flow around the Antarctic margin is mainly derived from shallower oceanographic data, but we suggest the ocean reanalysis product provides the best estimate available. As most available $\varepsilon_{Nd}$ values for seafloor surface

sediments around the Antarctic margin are based on measurements of the <63 μm grain size fraction, TASP assumes that a current velocity in the range of ~0.12-0.18 m/s is needed to start the resuspension and redistribution of the finest sediment particles (McCave and Hall, 2006; Gross and Williams, 1991). This threshold can vary significantly depending on properties such as the mean grain size, sorting and cohesivity of the sediment, but tuning this parameter within this range suggested a best match with seafloor surface sediments

when a threshold of 0.16 m/s is used (Section 2.7.1).

The ocean reanalysis velocity data are unlikely to capture the full variability in bottom current dynamics, which include significant changes in current speed and orientation on timescales as short as hours (e.g. Camerlenghi et al., 1997; Giorgetti et al., 2003). This poses the question as to whether the sedimentary record reflects predominantly an integrated mean of long-term bottom-current flow variability, or whether it is dominated by

episodes of peak current speed and current direction at these times. Here, it is assumed the latter case applies and, to account for the temporal variability in bottom current strength, peak flow velocities are assumed to be 2.5 times the monthly mean in the ocean reanalysis product. This relationship is based on measurements of bottom current strength at mooring sites around a contourite drift on the western Antarctic Peninsula continental rise (Camerlenghi et al., 1997; Giorgetti et al., 2003), which record peak velocities approximately two to three

times greater than mean velocities. The bottom current speed record from the Antarctic Peninsula drift is also comparable with other bottom current records (Gross and Williams, 1991). It is therefore suggested that, in areas where mean flow velocity exceeds 0.064 m/s in the reanalysis product, the aforementioned 0.16 m/s threshold speed for the start of resuspension and winnowing is exceeded during times when currents are strongest.

Consequently, all areas where bottom current velocities exceed 0.064 m/s are used as source locations for

debris. Our approach is supported by the fact that the sortable silt mean size of seafloor surface sediment from





another Antarctic Peninsula drift, recovered at a comparable water depth as the mooring measurements, suggests formation under a bottom current with a flow velocity matching the peak speed recorded in the mooring data (Hillenbrand et al., 2021). The extent of areas influenced by bottom current transport observed when using this threshold is viewed as realistic (see Fig. 10c). Detritus entrained in bottom currents is again routed using the MATLAB 'streamline' function.

As suspended particles will not be deposited uniformly over a given streamline, deposition over a streamline must be approximated. Although detailed modelling of bottom current erosion, transport and deposition is beyond the scope of this study, an approximation produces a realistic exponential decay in suspended particle concentration ($c$) over time ($t_s$, seconds):

$$c = c_0 \exp\left(-\frac{p t_s v_s}{y}\right) \tag{6}$$

Here, $p$ is the probability of a particle sticking to the ocean floor, $v_s$ is settling velocity, and $y$ is the thickness of the layer with suspension load in metres (Einstein and Krone, 1962). As we are not concerned with absolute amounts of sediment, the initial particle concentration ($c_0$) is irrelevant and set to 1. A value of $3.3\times10^{-4}$ m/s is used for the settling velocity, which is within the range expected for fine-grained detritus (Einstein and Krone, 1962; McCave, 2005). Suspended particle layer thickness ($y$) is highly variable and uncertain and was therefore tuned against seafloor surface sediment data, with a value of 15 m selected (Section 2.7.3).

The probability of the particle sticking to the bed ($p$) in Equation 6 can be calculated from bottom shear stress, $\tau_w$ (N m$^{-2}$), using a critical depositional stress, $\tau_c$ (N m$^{-2}$):

$$p = 1 - \frac{\tau_w}{\tau_c} \tag{7}$$

(Einstein and Krone 1962). $\tau_c$ is set to 0.08 N m$^{-2}$, in line with observed estimates for different classes of sediment which vary between 0.05 and 0.1 N m$^{-2}$ (Shi et al., 2015; Lumborg, 2005; McCave, 2008). In turn, $\tau_w$ can be approximated as a product of vertically averaged velocity ($\bar{U}$) and water density ($\rho$) using a quadratic law (Mofjeld, 1988; Garcia and Parker, 1993):

$$\tau_w = \rho C_d \bar{U}^2 \tag{8}$$

At the deep-sea water depths of interest here, the drag coefficient, $C_d$, can be estimated using:

$$C_d = \kappa^2 / [\log\left(\frac{H}{z_0}\right) - 1]^2 \tag{9}$$

Where $H$ is water depth and $z_0$ is the roughness length. Here we use $5\times10^{-4}$ m for the roughness length, which is a reasonable approximation given that the drag coefficient is insensitive to relatively small changes in water depth in the deep-sea (i.e., $H \gg z_0$) (Mofjeld, 1988). $\kappa$ is the von Kármán constant, 0.41. This approximation yields a spatially-variable drag coefficient in the order of ~$10^{-3}$, which is a magnitude consistent with observations (Umlauf and Arenborg, 2009). For simplicity, the basal current velocity is assumed to be approximately equal to the depth-averaged velocity.

To set the initial ε$_{Nd}$ composition of detritus mobilised by bottom currents, we use the output from the surface current method. To account for any seasonal changes in bottom current velocity, we then iterate the current





tracing for 24 months (2018-2019 ocean reanalysis product), using the output from the previous month to set the $\varepsilon_{Nd}$ value of sediment eroded where possible. Similar to the surface current estimate, the results for detritus moved by bottom currents are interpolated for 40 km around trajectories to account for unresolved pathways and for the fact that there is relatively little data to constrain modelled bottom current velocities in the ocean reanalysis product.

Similar to the surface current method (Section 2.3), this approach to bottom current transport assumes the transport along a given monthly mean trajectory occurs under steady-state conditions and does not account for changing currents during the duration of a detrital particle being suspended. This approach is by necessity a major simplification of the treatment of particle transport and sediment remobilisation by bottom currents. Nevertheless, the areas subject to sediment remobilisation are captured, alongside the provenance of detritus

reaching a given location through bottom current transport.

## 2.5 Gravitational downslope transport

Substantial volumes of glacimarine detritus can be moved through gravity-driven downslope processes, such as turbidity currents, slumps and debris flows. To represent these transport mechanisms, the mean $\varepsilon_{Nd}$ values from the surface and bottom current methods at all locations with a slope of >1.2°, mostly confined to the continental

slope, are selected.  This slope threshold was selected by tuning within a range of feasible values (Stow, 1994; Section 2.7.1). These grid cells are used as start locations from which the direction of gravitational transport is calculated by selecting the adjacent cell (including diagonals) with the lowest bed elevation. This next cell is set to the same $\varepsilon_{Nd}$ value as the starting cell. This iterates until a cell is repeated (i.e., an upwards slope is reached). TASP makes no attempt to account for variations in debris load over travel distance along a gravitational

transport pathway. This is deemed reasonable, as unlike ocean currents, such paths are unlikely to cross as the direction of travel will usually be approximately perpendicular to the coast (on the shelf) or the shelf break (on the continental slope). Thus, calculation of the relative quantity of sediment is unnecessary.

Bathymetric features not resolved at our 10 km model resolution mean gravitational flows are likely to carry detritus beyond the constraints of the approximate gravity flow paths. The gravitational transport $\varepsilon_{Nd}$ estimate is

therefore interpolated. Gravity flows do not necessarily shed their load once the slope falls below a threshold; turbidity currents, for example, can carry particles over 1000 kilometres (e.g. Mulder, 2011). As our gravity flow paths were already typically 300-600 km long, we interpolate a further 300 km at all locations beyond the shelf break.

Theoretically, the relief on the continental shelf can be sufficient in places for gravity flows to redistribute

sediment (e.g. Anderson et al., 1983; Hillenbrand et al., 2005). However, these processes are unlikely to transport significant amounts of sediment at the spatial scales (10's-100's km) considered here, and recent gravitational downslope deposits have only been identified locally on the very rugged, over-deepened inner continental shelf (e.g., Smith et al., 2009; Hogan et al., 2020). Thus, we do not interpolate the gravitational transport $\varepsilon_{Nd}$ estimate on the continental shelf, defined as water depths <1200 m. Although this depth threshold

is significantly deeper than the true average water depth of the shelf, which varies predominantly between ~400 and 500 m, using 1200 m avoids inclusion of most over-deepened areas on the innermost shelf close to the





present ice sheet margin, where interpolation should be avoided. Very few sediment core sites exist on the upper continental slope, so using this depth threshold has little impact on results.

Turbidity currents flowing perpendicular to the shelf break down the continental slope can carry suspended particles to locations on the continental rise where they are captured by bottom currents flowing parallel to the margin, i.e., approximately parallel to the shelf break (e.g., Rodrigues et al., 2022a; Hillenbrand et al., 2021). To account for this interaction between gravitational downslope processes and bottom current transport, the output of the gravity flow method was iterated once with the bottom current method described above. This improves the match with observed surface sediment $\varepsilon_{Nd}$ values. Although this approach does not explicitly represent

suspended material supplied by gravitational processes which is subsequently captured by bottom currents, using an iterative method does account for particle transport both parallel and perpendicular to the continental margin in both the bottom current and gravitational $\varepsilon_{Nd}$ value estimates.

### 2.6 Relative contribution of transport mechanisms

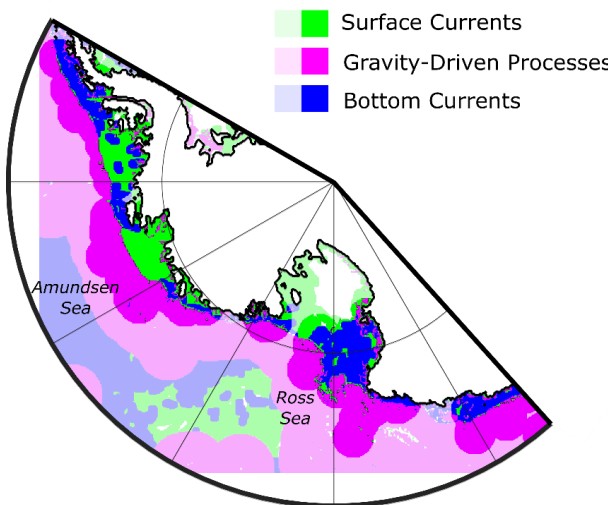

**Figure 6. Transport mechanisms for glacimarine detritus selected for the best estimate $\varepsilon_{Nd}$ value map. Colours are intense within 200 km of seafloor surface sample $\varepsilon_{Nd}$ constraints, and pale away from these constraints. White areas indicate grounded ice or a lack of sedimentation. Small black areas away from the grounding line denote where two or more methods predict an identical $\varepsilon_{Nd}$ value.**

To produce a 'best estimate' seafloor $\varepsilon_{Nd}$ map for comparison to data measured on surface sediments, it is

necessary to apportion the relative contribution of the three marine transport mechanisms (Sections 2.3-2.5) around the Antarctic continental margin. The TASP 'best estimate' map uses the value for gravity flows where they are present, bottom currents where gravity flows are absent, and surface currents where there is neither gravity flow nor bottom current estimate (Fig. 6). Assigning one method to each grid cell is not physically accurate, as all marine transport mechanisms may interact at a given location. However, TASP accounts for this

as it uses the surface current estimate as the source of the bottom current estimate, and the bottom current and surface current estimate maps as the source for the gravity flow estimate. A major strength of this approach is


that it does not require an estimate of the absolute quantity of detritus transported by each mechanism, which is useful given that the rates of these processes are very poorly constrained.

In some settings, marine transport processes can be estimated given observations of geomorphological features
(e.g., contourite drifts, channels eroded by turbidity currents), grain size parameters (e.g., content of coarse-grained clasts indicative of IBRD supply, percentage and mean of sortable silt as a proxy for bottom current vigour) and sedimentary structures (e.g., normally graded turbidites, winnowed layers of residual coarse-grained sediments). However, making these inferences requires good coverage with high-resolution bathymetric and seismic data and the collection of numerous sediment cores from targeted locations. Even then, however, it often
remains difficult to determine the fraction of the sediment transported by different mechanisms (e.g., Rodrigues et al., 2022a, 2022b). The approach used by TASP is therefore considered a necessary and appropriate simplification.

### 2.7 Parameter tuning and sensitivity tests

The methods described above provide a simplified way to estimate the provenance of debris generated beneath
the ice and transported in the ocean. As this is a highly complex - and often poorly understood – series of systems, there are some parameters which required tuning (Table 1). To explore the sensitivity of TASP to the most important of these parameters and select appropriate values, we tuned these by comparing the TASP prediction to seafloor surface sediments and computing the root mean square error (RMSE) and coefficient of determination.





**Table 1. Parameters used in TASP.**

| Parameter | Symbol | Unit | Value (if constant) |
|---|---|---|---|
| Erosion rate | $E$ | mm/yr | - |
| Ice basal shear stress | $\tau_b$ | Pa | - |
| Ice basal velocity magnitude | $u_b$ | m/yr | - |
| 'Quarrying' coefficient | $k$ | 1/Pa | $5.1\times10^{-10}$ |
| Debris concentration | $D$ | - | - |
| Debris yield above basal ice | $D_e$ | - | $10^{-5}$ |
| Initial debris concentration | $D_0$ | - | 1 |
| Basal ice thickness | $Z$ | m | - |
| Initial basal ice thickness | $Z_0$ | m | 4 |
| Iceberg basal melt rate | $M_b$ | m/day | - |
| Ocean Temperature | $T_s$ | °C | - |
| Magnitude of surface current velocity | $v_{mag}$ | m/day | - |
| Distance travelled since last cell | $d$ | km | - |
| Suspended sediment concentration | $c$ | - | - |
| Initial suspended sediment concentration | $c_0$ | - | 1 |
| Probability of particle sticking to bed | $p$ | - | - |
| Time | $t_s$ | second | - |
| Settling velocity | $v_s$ | m/s | $3.3\times10^{-3}$ |
| Suspended sediment layer thickness | $y$ | m | 15 |
| Bottom shear stress | $\tau_w$ | Pascal | - |
| Critical depositional stress | $\tau_c$ | Pascal | 0.08 |
| Seawater density | $\rho$ | kg/m³ | 1025 |
| Drag coefficient | $C_d$ | - | - |
| Vertically averaged velocity | $\bar{U}$ | m/s | - |
| von Kármán constant | $\kappa$ | - | 0.41 |
| Water depth | $H$ | m | - |
| Roughness length | $z_0$ | m | $5\times10^{-4}$ |
| Sediment decay coefficient | $\lambda$ | - | 5 |
| Gravity flow slope threshold | - | Degree | 1.0 |
| Gravity flow depth threshold | - | m | -1200 |

### 2.7.1 Sediment transport method thresholds

The sediment transport method selected at a given location will be influenced by the thresholds used to determine where transport mechanisms are active; in other words, the slope threshold for gravity flows and

velocity threshold at which bottom currents can transport sediment. Given this importance, these parameters were varied over a range of feasible values (Figs. 7 a,b). These results show that a velocity threshold of 0.16 m/s and slope threshold of 1.0 degrees produce the optimal results.



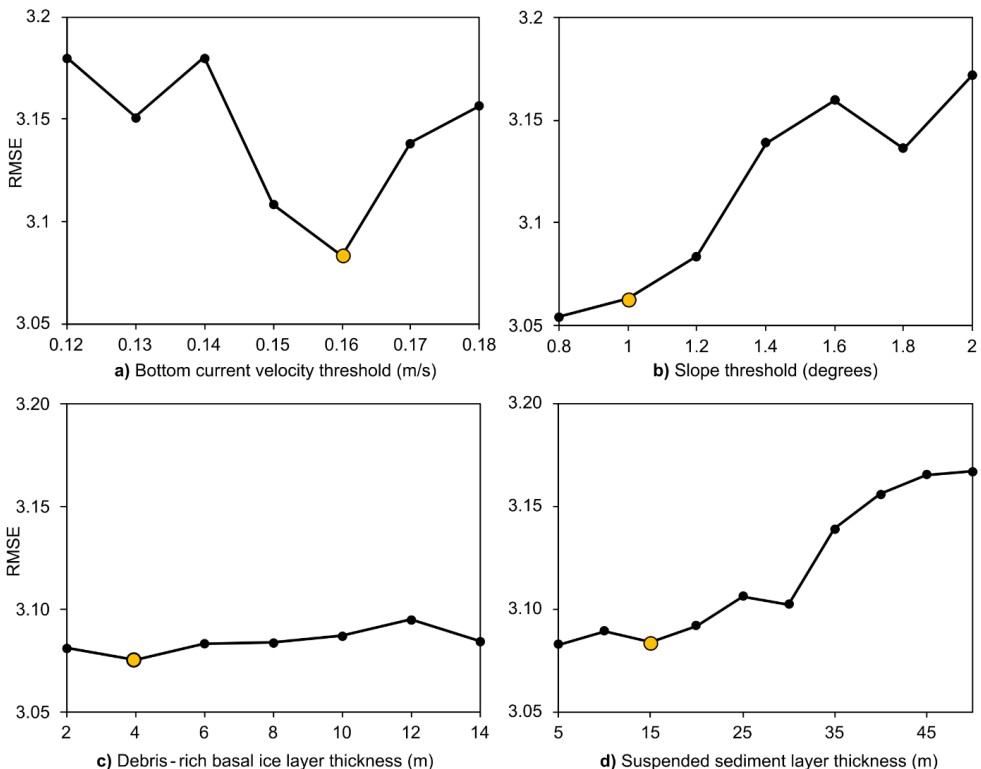

**Figure 7. Best estimate Root Mean Square Error (RMSE) results of tuning a) bottom current erosion rate threshold,**
**b) slope threshold, c) debris-rich basal ice layer thickness, and d) suspended sediment layer thickness parameters.**
**The parameter value selected is highlighted in yellow. Experiments were conducted using initial assumptions of a 1.2**
**degree slope threshold, 0.16 m/s bottom current velocity threshold, 6 m basal ice thickness and 15 m suspended**
**sediment layer thickness.**

### 2.7.2 Debris distribution in the ice column

As discussed in Section 2.3.1, debris distribution in the ice column will impact the distance this detritus is
transported in the ocean. Other options beyond a linear decay in distance from the bed (Equation 2) were
therefore explored. These included an exponential decline in debris yield:

$$D = \max\left(D_e, D_0\left(-\lambda\frac{Z}{Z_0}\right)\right) \tag{10}$$

Where a decay coefficient $\lambda = 5$ was selected for rain-out of basal debris over a length of time which agrees
closely with more in-depth studies of iceberg rafting (e.g. Hopwood et al., 2019). Experiments were also
performed with constant basal debris yield from the basal ice (i.e. $D = D_0$ while $Z < Z_0$). However, these
changes to debris distribution within the ice had a negligible impact on results.

Experiments were also run with varying thicknesses of the debris-rich basal ice layer (Fig. 7c). This parameter
was varied between 2 and 14 metres, in line with the observed range of thicknesses (Gow et al., 1979;
Christoffersen et al., 2010; Shaw et al., 2011; Pettit et al., 2014) and results compared with seafloor surface





sediment data. This revealed that the overall result is relatively insensitive to the chosen basal debris-rich ice layer thickness, with an optimal value of 4 m (Fig. 7c).

The amount of englacial debris relative to the basal debris concentration is another poorly constrained parameter. It was by default set to $10^{-5}$ times the maximum debris load at the base of the ice column based on

Dowdeswell and Murray (1990), but experiments were conducted varying it between $10^{-6}$ and $10^{-3}$. These results revealed TASP is insensitive to this parameter, although excluding this englacial debris entirely (i.e., setting the relative concentration to 0) does have a significant impact on the results (Fig. 4).

### 2.7.3 Bottom current suspended particle layer thickness

The treatment of bottom currents described in Section 2.4 requires estimates of several parameters, including the

bottom current threshold required to resuspend sediment, the relationship between mean and peak bottom current velocities, critical depositional stress and roughness length. Most of these parameters can be estimated based on theory and/or empirical data, but the suspended particle layer thickness is subject to significant uncertainty. This is because particles are unlikely to exist as a layer with a clearly defined upper surface. The thickness of the nepheloid layer on the Antarctic margin varies between less than 10 m and ~$10^2 – 10^3$ m (e.g.

Tucholke, 1977; Gilbert et al., 1998; Gardner et al., 2018). This parameter was therefore tuned against seafloor surface sediment $\varepsilon_{Nd}$ values measurements (Fig. 7d). Note that this effectively tunes the other constants used in Equations 6-9.

Very thin suspended particle layer thicknesses (≤5 m) led to very short transport distances, which are unlikely to capture bottom current transport. Using an increasingly thick suspended layer thickness resulted in transport

over longer distances, but a poorer match with seafloor surface sediment measurements (i.e. higher RMSE; Fig. 7d). A 15 m layer thickness was therefore used as a compromise, achieving a realistic transport distance without significantly worsening the match with seafloor surface sediment measurements. Although 15 m is less than the thickness of the nepheloid layer in several sectors of the Antarctic margin (Gardner et al., 2018), we argue it is physically plausible given that nepheloid layer thicknesses there have been measured in austral spring-summer,

when the nepheloid layer probably contained remnants of plankton blooms that took place just prior to these measurements. Furthermore, suspended particle concentrations are not uniform and are highest nearest the seafloor, and we are only concerned with bottom currents, which only comprise the lowermost part of the water column. Transmissometer data also suggest that suspended sediment concentrations increase markedly in the lower tens of metres of the water column (Tucholke, 1977; Gilbert et al., 1998; Gardner et al., 2020).





## 3. Results

### 3.1 Idealized basins

To evaluate whether TASP produces reasonable results, we use an idealised $\varepsilon_{Nd}$ value map for the IMBIE drainage basins (Zwally et al., 2012). Each basin was set to a value of 1 and elsewhere set to 0, and this repeated for five major drainage sectors in Antarctica (a) the Siple Coast, b) the Amundsen Sea, c) the central Transantarctic Mountains, d) Victoria/Oates/George V Land and e) the Antarctic Peninsula) to produce maps of the fraction of sediment derived from each basin (Fig. 8).

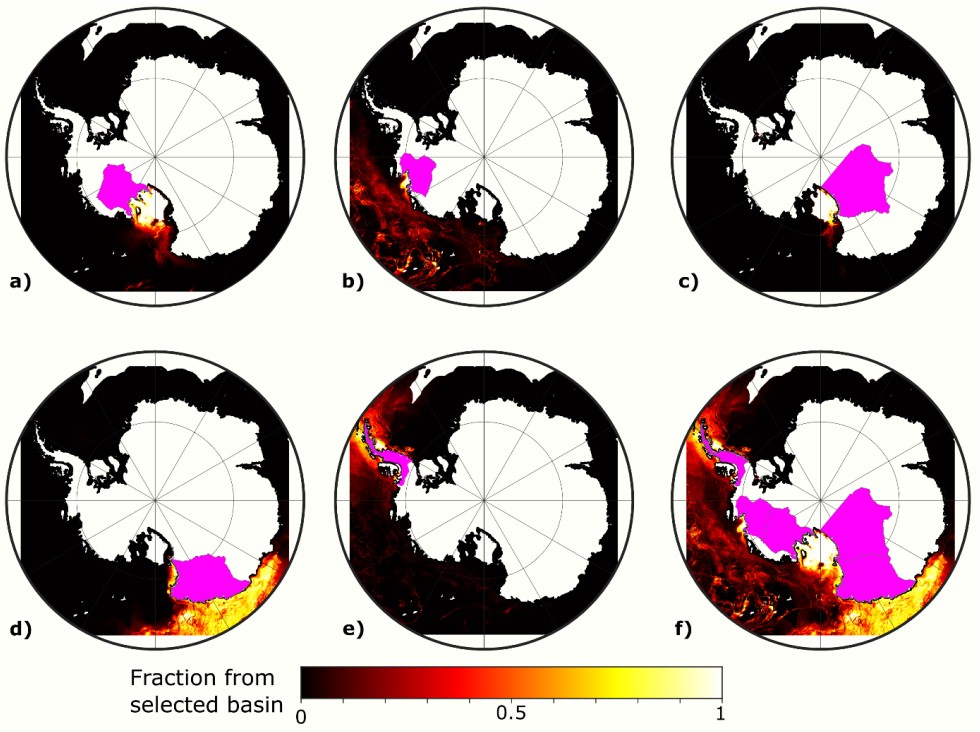

**Figure 8. The predicted fraction of offshore sediment originating from five IMBIE drainage basins shown in pink (a-e) and the total fraction from all 5 basins (f).**

This produces a realistic-looking distribution, indicating that TASP produces reasonable results (Fig. 8). This also suggests that different drainage basins provide detritus over quite different areas; whilst the Antarctic Peninsula and Amundsen Sea sectors contribute sediment over relatively large areas, the debris from the Ross Sea basins is deposited relatively locally.



**3.2 Comparison to seafloor surface sediment measurements**

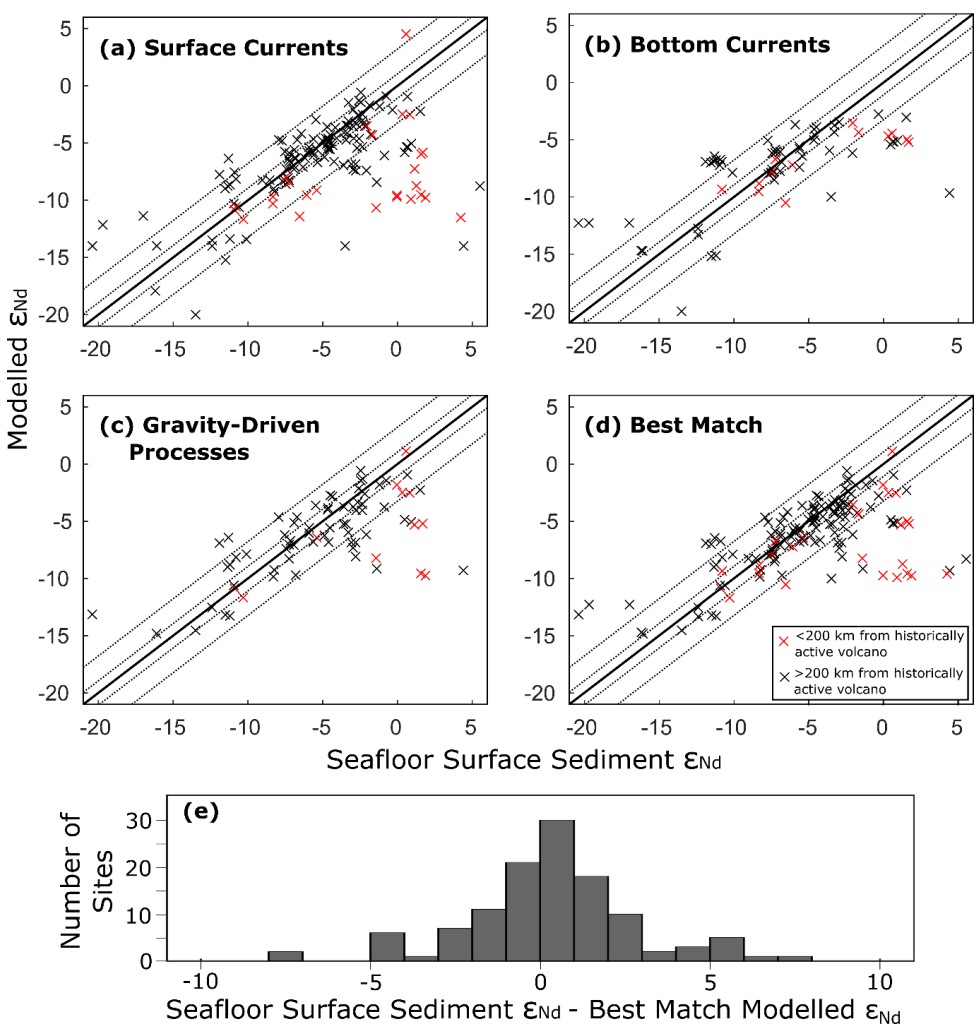


**Figure 9. Comparison between observed and modelled ε$_{Nd}$ values in seafloor surface sediments. Results are shown individually for each transport mechanism of glaciogenic detritus in the marine realm (a-c), plus selection of the nearest matching method (d). The solid 1:1 line is flanked by dotted lines indicating 1 and 3 epsilon unit deviations, respectively. Red crosses mark samples within 200 km of volcanoes which have been active in the historical**

**observational era; these samples may (or may not) be influenced by volcanic material with a high (~+5) ε$_{Nd}$ value. The difference between the measured and modelled (best match) ε$_{Nd}$ values of seafloor surface sediments (d) is also shown as a histogram (e).**



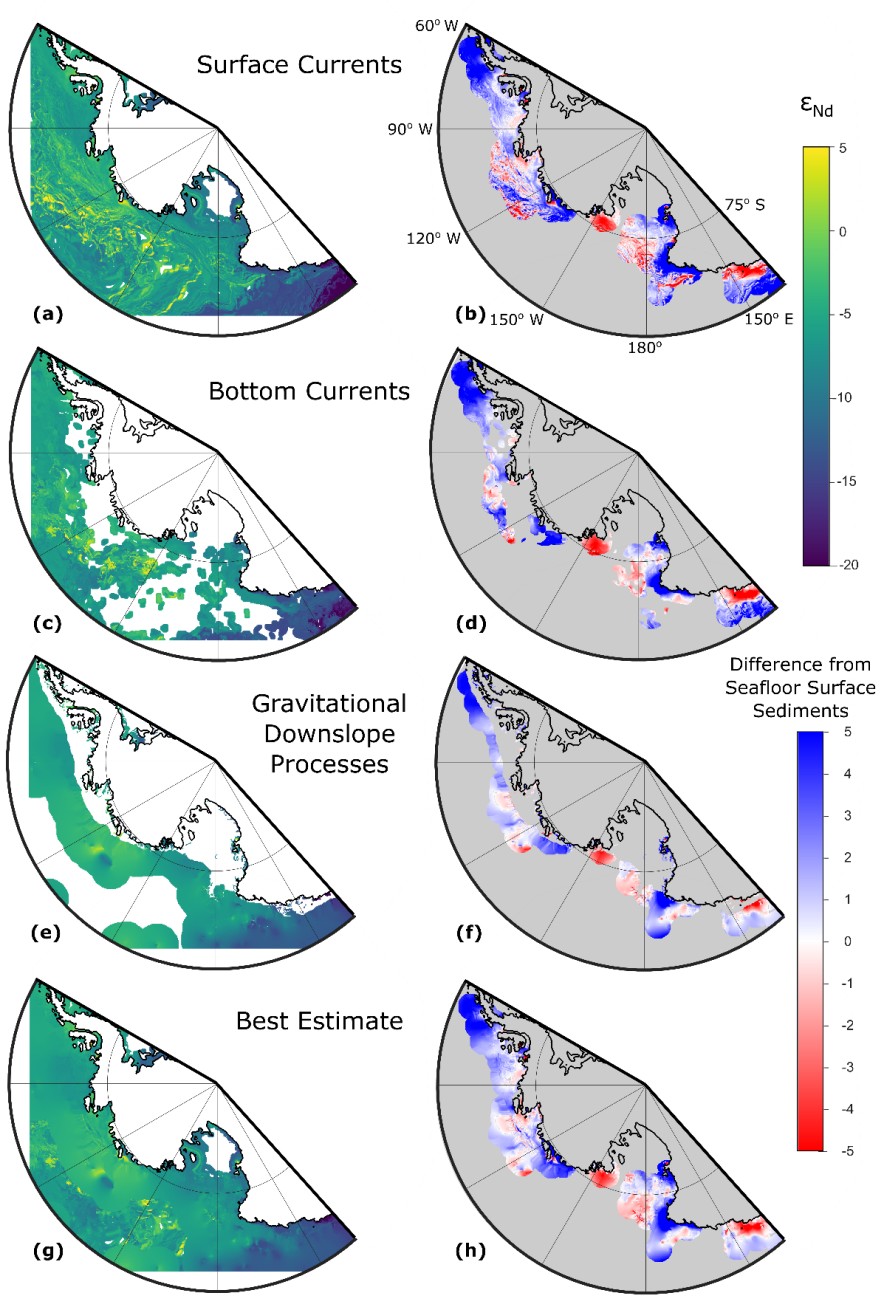

**Figure 10. Simulated ε_Nd values around the West Antarctic margin including the absolute values (a, c, e, g) and difference from the seafloor surface sediment data interpolation (b, d, f, h). The predicted ε_Nd provenance of seafloor surface sediments by modelled surface current (a & b), bottom current (c & d) and gravitational downslope (e & f) transport are shown, as well as the result when the closest matching method (surface current, bottom current, gravitational) to the interpolated core top ε_Nd values is selected for each grid cell (g and h).**





As well as the realistic distribution of sediment demonstrated by the idealised provenance signature maps (Fig.

8), TASP's results can be compared to the observed provenance of seafloor surface sediments to evaluate the

accuracy of the algorithm (Figs. 9, 10). Near most of the surface sediment sample locations, we produce a close

agreement between model results and measured data, with an overall RMSE of 3.05 and $R^2$ of 0.580 (Fig. 9).

TASP's predicted $\varepsilon_{Nd}$ values match 43% of surface sediment values within 1 epsilon unit and 81% within three

epsilon units. The $\varepsilon_{Nd}$ signals in published sediment provenance records off East Antarctica and in the Ross Sea

exceed three epsilon units (e.g. Cook et al., 2013; Wilson et al., 2018; Marschalek et al., 2021). This implies our

model predictions are sufficiently accurate at most sites to exceed the likely amplitude of changes in $\varepsilon_{Nd}$ values

caused by a change in the ice sheet drainage pattern or in areas of subglacial erosion and thus grounded ice-sheet

extent. TASP can, therefore, reproduce observed seafloor surface sediment $\varepsilon_{Nd}$ values to a sufficient degree of

accuracy to provide useful predictions of past ice sheet configurations.

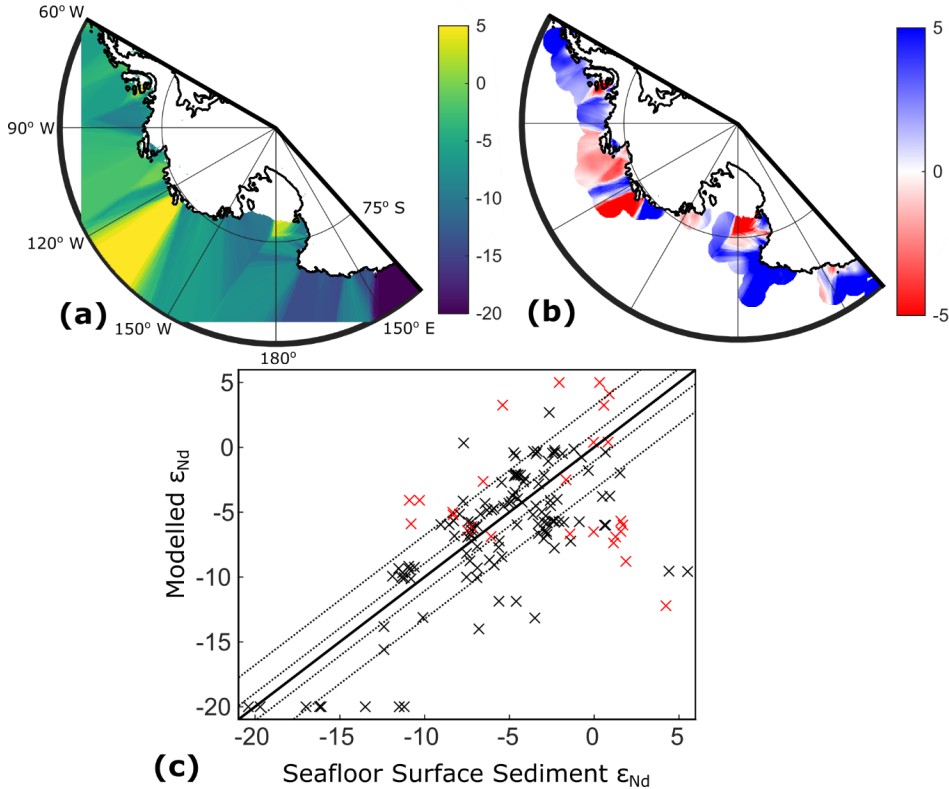


**Figure 11. Results when using inverse distance weighting from the ice sheet margin for predicting $\varepsilon_{Nd}$ values of seafloor surface sediments, including a) the extrapolated $\varepsilon_{Nd}$ values and b) the difference between $\varepsilon_{Nd}$ values predicted by this method and those measured on the surface sediments. c) Shows the scattered relationship between measured and modelled seafloor surface sediments.**

To investigate TASP's performance compared to assuming that debris is simply advected approximately

perpendicular to the coast, an inverse distance weighting interpolation was used to extrapolate the provenance

signature of the glaciogenic detritus supplied to the ice sheet margin offshore (Fig. 11). This resulted in a RMSE



of 3.77, considerably worse than TASP's prediction incorporating marine processes (RMSE = 3.05). The algorithm's incorporation of marine detrital particle transport mechanisms therefore results in a much closer match to $\varepsilon_{Nd}$ values at seafloor sample sites. This is because an inverse distance weighting method does not reflect transport of detritus approximately parallel to the coast by ocean currents/icebergs.

Regionally, model-data match is particularly good, and nearly always deviates by less than one $\varepsilon_{Nd}$ unit, in the central Ross Sea and the Bellingshausen Sea (Fig. 10). Performance of TASP is also better in deep waters beyond the continental shelf. Furthermore, the Amundsen Sea embayment offers a dense offshore sample network to compare TASP's results to (Simões Pereira et al., 2018; 2020; Fig. 2). The modelled $\varepsilon_{Nd}$ values do not resolve finer-scale provenance features, but the match is good close to the terminus of both Pine Island and Thwaites glaciers and in the central Pine Island-Thwaites paleo-ice stream trough (Fig. 10). TASP also broadly captures the mixing of the distinct detritus supplied by each of these two major WAIS outlets (Simões Pereira et al., 2020).

## 4. Discussion

### 4.1 Distribution of different sediment transport processes

On parts of the continental shelf, low bottom current velocities mean the surface current estimate often provides the best estimate $\varepsilon_{Nd}$ values (Fig. 6). The results for the Amundsen Sea and eastern Ross Sea continental shelf suggest a particularly strong influence of surface currents/IBRD on the provenance of the sediments deposited in these regions. The surface current estimate is also used in the best estimate in some areas further offshore. However, in these regions, individual iceberg drift trajectories driven by surface current flow can be distinguished in the $\varepsilon_{Nd}$ output. This reflects icebergs being less common, meaning our result often reflects a single iceberg trajectory rather than a mean of many drift paths as is the case closer to the continent. The strong, consistent imprint on iceberg drift of the Antarctic Circumpolar Current (ACC) north of its southern boundary (SBACC; Fig 1) means different icebergs reaching a location north of the SBACC are likely to have a similar source; thus, there is a reduced need for multiple iceberg tracks over the same point. Despite this, a greater density of iceberg trajectories than that achieved by TASP is still required to reduce the gaps between the trajectories. The surface current method is therefore viewed as less reliable for predicting IBRD deposition in the deep ocean, where a surface current record spanning several decades and/or at a higher resolution would likely be required to get a denser network of iceberg trajectories over these areas, adding significant computational cost.

The threshold for bottom current influence on fine-grained sediment provenance is reached in many locations on the continental shelf and parts of the continental slope and rise (Fig. 6). These areas are particularly concentrated along the western Ross Sea continental shelf, Antarctic Peninsula, Marie Byrd Land coast, Victoria Land coast and George V Land coast. In the Ross Sea, this may reflect sediment transport along Ross Sea Bottom Water export pathways (Orsi and Wiederwohl, 2009).

Beyond the continental shelf, gravitational downslope processes dominate in nearly all locations (Fig. 6). Compared to the other sediment transport mechanisms, sediment core sites in these regions typically provide the closest match with TASP's predictions, with an RMSE of 3.20 (Figs. 9, 10). Substantial model-data mismatch



for the gravitational downslope mechanism is limited to sites with special, localised environmental settings, which are discussed in detail below (Section 4.3). The close agreement between TASP and the measured seafloor surface sediments suggests that sediment transport perpendicular to the shelf break is dominant further offshore, and that the amount of detritus moved by gravitational downslope processes typically exceeds that of detritus solely supplied by ocean currents in these areas.

**4.2 Volcanic detritus**

Wind-blown dust from other continents reaches Antarctica and could influence the provenance signature of marine sediments (Neff and Bertler, 2015). However, such sources are minor in most Antarctic continental margin settings used for sediment provenance studies (e.g., Walter et al. 2000; Wengler et al. 2019) and are therefore neglected.

However, volcanic material/tephra erupted above the ice surface and carried offshore, either by winds or in layers within icebergs, could be present in the seafloor surface sediments in areas close to and downwind of volcanoes. This detritus could substantially influence $\varepsilon_{Nd}$ values, as Cenozoic volcanic material in West Antarctica and Victoria Land is highly radiogenic (~+5 $\varepsilon_{Nd}$) and has a high Nd concentration (Goldich et al., 1975; Aviado et al., 2015; Futa and LeMasurier, 1983; Hart et al., 1997). TASP does not make any attempt to

account for the influence of this volcanic material. Indeed, areas where predicted $\varepsilon_{Nd}$ values that are less radiogenic (lower) than those observed include the coast of Victoria Land, part of the Hobbs Coast of Marie Byrd Land and the area around the northern tip of the Antarctic Peninsula (Fig. 10), and this mismatch is likely attributable to volcanoes in these areas that are either still active or have been active in the recent past (e.g., Dunbar et al., 2021; Patrick and Smellie, 2013; Fig. 2).

The explanation for mismatch in these areas is supported by the observation that clay-sized detritus in sediments offshore from the volcanic South Shetland Islands, near the northern tip of the Antarctic Peninsula, has a relatively high smectite content; Cenozoic volcanic material has a strong imprint on clay mineral assemblages and the detrital $\varepsilon_{Nd}$ signature continental rise sediments in the region (e.g., Hillenbrand and Ehrmann, 2005; Hillenbrand et al., 2021; Wang et al., 2022). Furthermore, grain size distributions and visual observations in

McMurdo Sound suggest that seafloor surface sediments there are dominated by (initially) airborne detritus that had been blown by strong winds both directly into the ocean and onto sea ice and the McMurdo Ice Shelf before being deposited on the seabed (Kellogg et al., 1990; Atkins and Dunbar, 2009). The observed model-data mismatches in the vicinity of volcanoes show that marine sediment provenance records using $\varepsilon_{Nd}$ values are likely influenced by radiogenic volcanic material; Nd isotope records from these areas should be interpreted

with caution.

Three samples collected adjacent to the Adare and Hallett peninsulas on the Northern Victoria Land coast recorded particularly radiogenic $\varepsilon_{Nd}$ values which are also not captured by TASP (outliers in Fig. 9; Table A2). Visual mineralogical composition and $\varepsilon_{Nd}$ values of >+4 suggest they consist almost entirely of detritus derived from the late Cenozoic volcanic rocks constituting these peninsulas. However, as the source volcanoes have not

been active for at least ~2 million years (LeMasurier et al. 1990), this discrepancy cannot be attributed to recent tephra deposition but must result from the supply of locally eroded volcanic material. These results instead highlight a limitation of our 10 km model resolution, which is not able to resolve such localised areas of a



particular rock type (Fig. 2). If such sites close to the shore were of interest for sediment provenance tracing, a more localised, high resolution modelling approach would need to be pursued to yield more accurate results.

To visualise the locations of seafloor surface sediment samples which are in the vicinity of recently active volcanoes and therefore potentially influenced by significant volcanic deposition, sites within 200 km are highlighted in red in our scatter plots and excluded from our statistics (Fig. 9). We emphasise that the exclusion of these sites does not imply that sediment provenance data here are not useful; instead, the provenance data in these areas are viewed as having the potential for unaccounted-for volcanic influences. One exception is the

recent volcanic activity suggested in the Hudson Mountains near Pine Island Bay in the eastern Amundsen Sea embayment (Patrick and Smellie, 2013; Corr and Vaughan, 2008), which we do not use to exclude data. Any volcanic input to marine sediments here is likely very local and minor given both (i) the restricted geographical distribution of this tephra on land (Corr and Vaughan, 2008) and the rare and locally restricted occurrence of macroscopic tephra in marine sediments from the area (Herbert et al., 2023), and (ii) the large supply of non-

volcanic glaciogenic debris from the adjacent Pine Island and Thwaites glaciers. For example, a petrographic analysis of coarse clasts >2 mm recovered in a box core in Pine Island Bay downstream of these two ice streams revealed that volcanic rocks accounted for only 0.5% of all the lithologies present (Lindow et al., 2016).

Volcanogenic detritus could also reach sample sites from volcanic seamounts and islands (Fig. 2b). These range in age from the early Cenozoic to recent and include the Marie Byrd Land Seamounts in the Amundsen Sea

(Kipf et al., 2014); Peter I. Island (Prestvik and Duncan, 1991) and the DeGerlache Seamounts (Hagen et al., 1998; Hagedorn et al., 2007) in the Bellingshausen Sea; islands north of Victoria Land (Johnson et al., 1982); and islands and seamounts in the western Ross Sea (Lawver et al., 2012; Rilling et al., 2009; Panter and Castillo, 2007). Radiogenic isotope compositions are similar to the aforementioned Cenozoic volcanic rocks (Kipf et al., 2014; Panter and Castillo, 2007; Prestvik and Duncan, 1991). These seamounts and islands are not accounted for

in TASP's predicted $\varepsilon_{Nd}$ values for seafloor surface sediments as the extent and magnitude of the influence of such features is uncertain.

**4.3 Localised impact of grounded icebergs and sea ice**

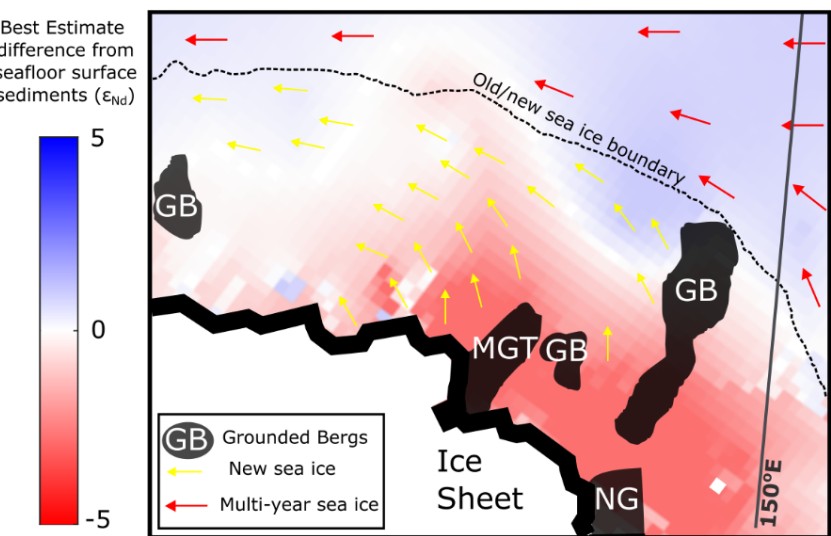

**Figure 12. Close up of the Mertz Glacier Tongue (MGT) region imaged in June 1999, adapted from Massom et al. (2001) and Marsland et al. (2004). Location of map shown in Appendix Fig. A1. Red-blue colours indicate the discrepancy between the modelled and core top $\varepsilon_{Nd}$ values as in Fig. 10h. Overlain is a map of old (multi-year) and new sea ice trajectories (red and yellow arrows, respectively), which are deflected to the north due to grounded bergs (GB) and fast ice. Note that the MGT had undergone a major calving event in 2010 but since then another seaward extending tongue consisting of floating glacial ice protruding from Ninnis Glacier (NG) and Mertz Glacier and amalgamated by fast ice has established (e.g., Wang et al., 2018).**

Offshore of the Wilkes Subglacial Basin and adjacent to the King Edward VII Peninsula, predicted $\varepsilon_{Nd}$ values are more radiogenic (higher) than those measured in the seabed sediments (Fig. 10). A key factor influencing $\varepsilon_{Nd}$ values in these regions is the supply of relatively radiogenic detritus via the westward flowing Antarctic Coastal Current, which in both these areas appears to be comprising a greater fraction of detritus in our model than in reality. This is a result of icebergs remaining closer to the coast in the simulation. In these areas, we suggest that the ocean reanalysis product may not be resolving local factors - such as local ocean currents and the presence of grounded icebergs and fast ice – which lead to more far-travelled icebergs carrying relatively radiogenic detritus being diverted north, away from the coast. Offshore from Wilkes Subglacial Basin, for example, high-resolution modelling of sea ice trajectories around the Mertz Glacier area shows that icebergs coming from the east are deflected north by a melange of fast ice and grounded icebergs (Marsland et al., 2004), allowing locally derived debris (with more negative $\varepsilon_{Nd}$ values) to be more dominant in the lee side of this "promontory" (Orejola et al., 2014). This area of deflection matches remarkably well with the area where TASP's results and observations differ (red and blue areas in Fig. 12). Similarly, a gyre offshore from King Edward VII Peninsula deflects icebergs, which drift within the Antarctic Coastal Current westwards along the Amundsen Sea shelf, to the north before injecting them into the eastward flowing ACC (Keys, 1990; Rackow et al., 2017). This implies that grounded bergs, sea ice and unresolved ocean currents are a key control on the provenance of continental





shelf sediments in these regions and should be carefully considered when interpreting sediment provenance records.

At some sites, observed mismatch may also be linked to uncertain dating of seafloor surface sediments. To achieve a good spatial coverage, we group all available measurements on samples likely to be late Holocene to modern in age, but difficulties in radiocarbon dating of often carbonate-free Antarctic sediments mean these samples may vary in age throughout the Holocene on a site-by-site basis.

**5. Conclusions and future direction**

We present TASP, the first algorithm which predicts offshore sediment provenance around Antarctica using the results of ice sheet modelling and approximations of marine detrital particle transport mechanisms. Comparison to seafloor surface sediments verifies that TASP produces useful results and helps understanding of the modern sedimentary system. For instance, our findings show that westward flowing currents control deposition on the continental shelf, whilst a good model-data match in deeper waters indicates gravity-driven processes are the

dominant sediment delivery mechanism there. Furthermore, our results suggest regions where local factors such as sea ice or recent volcanism are likely impacting present day sedimentation. Our study demonstrates the vital importance of considering ocean transport processes when interpreting sediment provenance records.

On the continental shelf, we capture the general pattern of $\varepsilon_{Nd}$ values observed around West Antarctica and the East Antarctic George V Land margin. Any sites with substantial model-data disagreement can readily be

explained by specific regional factors or geological uncertainties not captured in the provenance tracing. Furthermore, the TASP algorithm is particularly powerful beyond the continental shelf, where it produces a close match when using a simulation of the modern ice sheet to predict $\varepsilon_{Nd}$ values of seafloor surface sediments. This is likely because greater integration of provenance signals makes our approach less reliant on precise mapping of uncertain subglacial geology.

The development of the TASP algorithm will permit future application to palaeo ice sheet model simulations. These data will help interpret existing and new sediment provenance records by suggesting what ice sheet configurations could produce the changes in $\varepsilon_{Nd}$ values (and potentially other provenance proxies) recorded at any given core site. Tighter coupling of numerical ice sheet model simulations and provenance data offers the potential for the advancement of both fields; results of numerical modelling can be directly compared to real-

world constraints, and interpretations of geochemical data can be tested, visualised and quantified in terms of ice sheet volume and extent.

Improved knowledge of subglacial geology, particularly in much of the West Antarctic interior, would enable TASP to produce even more useful and accurate results. Future geophysical surveys, as well as drilling campaigns targeting subglacial sediments and bedrock, will reduce uncertainty in our $\varepsilon_{Nd}$ map and thus

predictions offshore. Additional insights could be achieved by expanding TASP to different parts of the Antarctic continent. Our method could also be adapted for numerous provenance proxies, such as detrital mineral dating, major and trace element geochemistry, clay and heavy mineralogy, or clast petrography. Each of these proxies offers a different sensitivity to changes to erosion patterns and thus different precision and insights in different regions and at different times in the past (Licht and Hemming, 2017).



**Code and Data Availability**


The TASP code is available under a GPLv3 licence. The version of TASP used to produce the results in this paper was run using MATLAB version R2022a and is archived on Zenodo (Marschalek, 2023; https://doi.org/10.5281/zenodo.11449956). An example ice sheet model output used to produce the results in this paper is also available in the TASP repository (DeConto et al., 2021). The sub-ice shelf melt rate data

(Adusumilli et al., 2020) can be found here: https://doi.org/10.6075/J04Q7SHT and the ORAS5 ocean reanalysis product files used (velocity and sea surface temperature) can be downloaded here: https://doi.org/10.24381/cds.67e8eeb7 (Copernicus Climate Data Store, 2021). As these are published datasets, it is not appropriate to copy these data in our TASP repository as it could lead to confusion about the original source of the data. However, the files are freely accessible in these archives or can be provided by the

corresponding author on request.

The new neodymium isotope data published here are available here: https://doi.org/10.5281/zenodo.7548284.

**Author Contributions**

JM, EG and TvdF designed the study. JM developed the code with assistance from EG. CDH and MS advised on sedimentological and glaciological aspects of the manuscript, respectively. LH collected the new Nd isotope

data to complement existing published datasets.

**Competing Interests**

The authors declare that they have no conflict of interest.

**Acknowledgements**

JM and TvdF acknowledge funding from the NERC (DTP scholarship to JM, NE/L002515/1, and grants

NE/R018219/1 and NE/W000172/1 to TvdF). The Ross Sea surface sediments, from which new Nd isotope data were collected, were supplied by Helen Bostock from NIWA, Wellington, New Zealand.





### Appendix 1 - Regional Description of the Neodymium Isotope Composition Map

#### A.1.1 Approach

Here we present a region-by-region summary of the sources used to define the extent and Nd isotope
        composition of the bedrock of West Antarctica and adjacent areas of East Antarctica. A description of
        geological formations, lithologies and the tectonic framework of the study area is not relevant to this study and
        so is not discussed here. For reviews of East Antarctic, West Antarctic and Transantarctic Mountain geology,
        see Boger (2011), Jordan et al. (2020) and Goodge (2020), respectively.

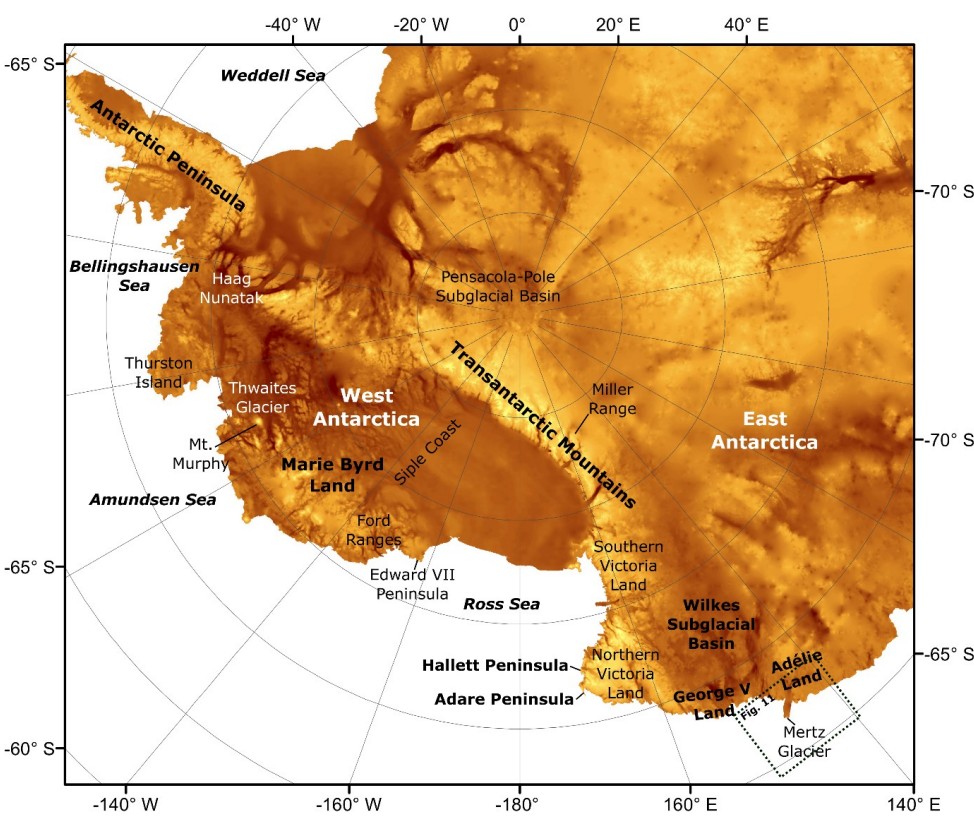


**Figure A1. Geographical locations mentioned in the supplemenary and main text displayed on BedMachine topography (Morlighem et al., 2020). The area covered by the map shown in Fig. 11 is marked by the box with the black dotted outline.**

        The $\varepsilon_{Nd}$ value map (Fig. 2a) was based on measurements of rock $\varepsilon_{Nd}$ values from published literature sources
which had been previously compiled (Simões Pereira et al., 2018, 2020; Marschalek et al., 2021). These datasets
        were extended across the domain used here. Where coordinates for the locations of rock samples were not given
        in the original publication, they were taken from the SCAR Antarctic Digital Database Map Viewer
        (https://www.add.scar.org/) based on the location name.





To convert these literature 'point data' into a continuous map, outcropping rocks were grouped. This was

facilitated by the recent digital geological mapping of Antarctica from the SCAR GeoMAP project (Cox et al.,

2023). To simplify this map, which comprised polygons at a resolution far beyond that required for this study,

adjacent outcrops were grouped where subglacial continuity between exposures was deemed reasonable (see

example in Fig. A2). For each major rock type, the point $\varepsilon_{Nd}$ values were then interpolated and masked to the

area of the simplified polygons. A resolution of 10 km was used, which is sufficient given the uncertainties

resulting from the ice cover and matches the resolution used in the ice sheet model. These areas of exposed rock

can be considered as having a high level of confidence in the final map, particularly when $\varepsilon_{Nd}$ values for the

outcropping rock group have been measured at numerous localities.

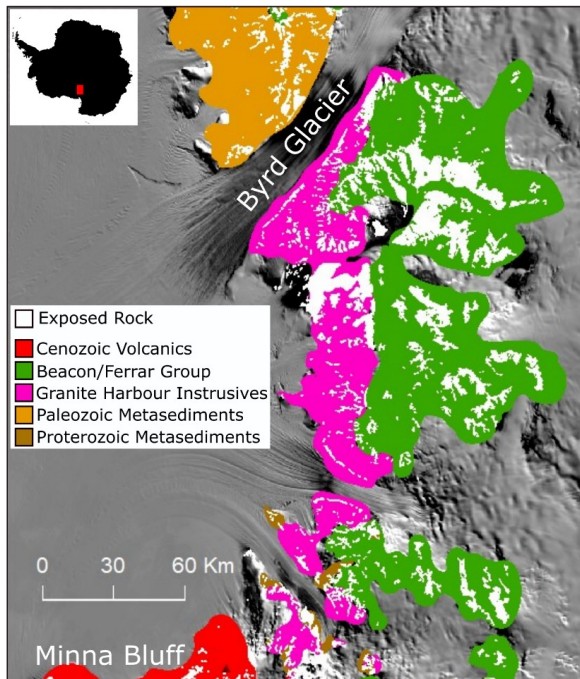

**Figure A2. Example of the high-resolution rock polygons from GeoMAP (white; Cox et al., 2023) versus the**

**simplified polygons used to construct the $\varepsilon_{Nd}$ value map (coloured). The latter are better suited to the >~10 km**

**resolution of interest here. The region shown is a section of the Transantarctic Mountains between Byrd Glacier and**

**Minna Bluff. The geology is overlain on MODIS imagery (Haran et al., 2014). The location within Antarctica is**

**indicated in the upper left inset.**

Most of the study area is covered by ice, making direct geological observations impossible. Pertinent airborne

and on-ice geophysical (i.e., gravity, magnetic, subglacial geomorphology obtained by radar) studies were

therefore used to inform $\varepsilon_{Nd}$ values beneath the ice by making interpretations regarding the subglacial geology

(e.g. Jordan et al., 2013a; Goodge and Finn, 2010; Studinger et al., 2006; van Wyck de Vries et al., 2018;

Paxman et al., 2019). In some regions, this approach can produce reasonably robust inferences. However, in

other areas, relationships between exposed and subglacial geology are more obscure. Even in areas where

geophysical constraints are good, the subglacial bedrock there may be hundreds (or thousands) of kilometres away from the nearest comparable outcropping rocks. Consequently, using interpolation from the nearest outcrops to assign $\varepsilon_{Nd}$ values, as described above for exposed rocks, could lead to bias towards only the geographically nearest measurement. Interpolation may therefore fail to reflect the true variability of the subglacial geology in such areas. To avoid this, areas where a subglacial rock type was predominantly

'geophysically'-defined were uniformly assigned a mean $\varepsilon_{Nd}$ value based on all suitable measurements of the rock type.

The sources used to determine the extent of subglacial geology and to estimate the $\varepsilon_{Nd}$ value for each rock group are discussed in detail below. To keep the initial offshore $\varepsilon_{Nd}$ value estimate as independent as possible, an inverse approach was specifically avoided when constructing the subglacial $\varepsilon_{Nd}$ map (i.e., using offshore values

to predict onshore ones), except in a few locations where virtually no geophysical or geological data were available. A formal inversion for subglacial geochemistry, similar to that recently applied to fluvial provenance (Lipp et al., 2020), would undoubtedly provide a much-improved match with modern observations as well as insights into subglacial geology to apply to simulations of past ice sheets. However, further development would be required to implement a formal inverse approach in a glacial setting, as additional information beyond an

offshore measurement near the ice sheet margin is required to give structure to interior $\varepsilon_{Nd}$ values and avoid uniform values along a flow line.

To indicate the approximate uncertainty in the map produced, areas are classified with confidence levels 1-4 (Fig. 2b):

1) Exposed rocks (high confidence; black) – exposed rock with $\varepsilon_{Nd}$ data. The quantity and spatial
distribution of $\varepsilon_{Nd}$ measurements varies between rock types and, in some instances, sparse data meant that the number of data points used in the interpolation across a rock group was low.

2) Geophysical constraints (moderate confidence; dark grey) – areas with subglacial geology constrained by gravimetric/magnetic/geomorphological etc. data. Epsilon Nd values inferred for rock groups are based on the nearest equivalent exposures, but $\varepsilon_{Nd}$ values may be different away from the rock outcrop
with $\varepsilon_{Nd}$ data, or certain rock groups may be over- or under-represented. For instance, in Marie Byrd Land outcrops of Neogene volcanic rocks dominate, but Palaeozoic basement and Cretaceous plutonic rocks are much more widespread (e.g. Rocchi et al., 2006).

3) Inverse/point constraints (low confidence; light grey) – areas with few direct constraints, limited to isolated subglacial samples obtained by drilling through the ice. Epsilon Nd values are determined from
these limited samples and surrounding rock exposures, loosely informed by offshore provenance data. This approach differs from a formal inversion as the offshore data are not used to directly predict $\varepsilon_{Nd}$ values beneath the ice; instead, they are used to infer likely dominance of outcropping rock groups in areas without any other constraints.

4) No estimate made (white) – these areas are without any rock type constraints, filled using Kriging.

Different rock types have different concentrations of Nd, meaning some rock types will contribute disproportionately to the $\varepsilon_{Nd}$ values of subglacial detritus. This effect is not accounted for here due to the uncertainty in Nd concentrations of different rock types, making quantitative assessments challenging. Assuming constant Nd concentrations still yields useful results, but the inclusion of variable concentrations in





future work would improve the accuracy of our results. Our approach also neglects any pre-glacial sedimentary

strata, marine and other sediments deposited when ice sheets were more retreated than today (e.g. under the WAIS, Wilkes-Adélie Land sectors of the East Antarctic Ice Sheet), or allochthonous subglacial till overlying in-situ bedrock at the base of grounded ice. However, subglacial mixing of detritus is to some degree accounted for by the interpolation of $\varepsilon_{Nd}$ values between regions with assigned $\varepsilon_{Nd}$ values, as well as assumptions of mean $\varepsilon_{Nd}$ values of the surrounding rock types in regions where pre-glacial/marine sedimentary strata or till sheets are

likely present under the ice. These simplifications have the effect of smoothing the bedrock $\varepsilon_{Nd}$ values. Fully accounting for reworking of subglacial sediments would require a full integration of $\varepsilon_{Nd}$ values in time-evolving ice sheet model runs, which is beyond the scope of this study.

Some regions – particularly the interior of the WAIS towards Marie Byrd Land, the southernmost Transantarctic Mountains and the hinterland of the Bellingshausen Sea – lack reliable geophysical constraints. In these regions,

initial estimates were made by simply interpolating the map produced over the remaining gaps. However, this led to some obvious discrepancies between the $\varepsilon_{Nd}$ values of rocks on land and coastal marine sediments. This effect was particularly pronounced in regions of West Antarctica, where geophysical bedrock studies typically focus on detecting comparatively radiogenic subglacial igneous bodies. The lack of constraint on surrounding rock biases $\varepsilon_{Nd}$ values. To account for this, most large areas of the study area with unassigned $\varepsilon_{Nd}$ values were

prescribed a value informed by sparse subglacial till samples (e.g. Farmer et al., 2006) and surrounding rock types, which markedly improved agreement with marine surface sediment $\varepsilon_{Nd}$ values. As spatial variability in $\varepsilon_{Nd}$ values is unknown in these regions, the assigned $\varepsilon_{Nd}$ values were assumed to be uniform.

### A.1.2 George V Land and the Wilkes Subglacial Basin

The extent of pre-Cambrian East Antarctic rocks is mapped according to Goodge and Finn (2010), which is

broadly consistent with ADMAP2 magnetic anomalies (Golynsky et al., 2018). The coastal extent of these pre-Cambrian rocks is based on interpretation of these data in Bertram et al. (2018). Limited measurements of this rock group give spatially variable $\varepsilon_{Nd}$ values, spanning approximately -33 to -11 in the Miller Range of the central Transantarctic Mountains (Borg et al., 1990; Borg and DePaulo, 1994), -24 to -22 along the Adélie Land coast (Peucat et al., 1999), and -16 to -15 at the bed of the EPICA Dome C ice core (Delmonte et al., 2004). The

pre-Cambrian rocks are assigned a uniform $\varepsilon_{Nd}$ value of -20, which is considered reasonable given the limited exposure of these rocks. Beyond the edge of the area studied, no attempt was made to compile literature $\varepsilon_{Nd}$ values as erosion rates are low (e.g. Jamieson et al., 2010; Fig. 3a) and the Antarctic Coastal Current will carry any detritus reaching the coast westwards out of the study area (e.g. Fig. 5). Rocks in other parts of Antarctica are thus unlikely to substantially impact the provenance signature of the regions studied offshore here, although

some ice flowlines do cross beyond the edge of the study area. Based on the assumptions described above, the rest of East Antarctica was also assigned a uniform $\varepsilon_{Nd}$ value of -20.

Exposures of the late Neoproterozoic to Ordovician Wilson, Bowers and Robertson Bay Terranes are widespread throughout Northern Victoria Land. For these three respective rock groups, measured $\varepsilon_{Nd}$ values range from -22.6 to -11.1 (Talarico et al., 1995; Henjes-Kunst and Schussler, 2003; Estrada et al., 2016), -19.9

to -10.5 (Tonarini and Rocchi, 1994; Henjes-Kunst and Schussler, 2003) and -15.6 to -12.3 (Henjes-Kunst and





Schussler, 2003). However, most measurements cluster around a mean $\varepsilon_{Nd}$ value of approximately -14, which is assigned uniformly to these groups.

Granite Harbour Intrusive rocks are exposed in Victoria Land and, although these have relatively uniform $\varepsilon_{Nd}$ values with a mean of ~-12 (Cox et al., 2000; Armienti et al., 1990; Dallai et al., 2003; Rocchi et al., 1998;

2009), their $\varepsilon_{Nd}$ values are allowed to vary spatially in our $\varepsilon_{Nd}$ map. These granitic rocks are also suggested to be present in the Wilkes Subglacial Basin, but the southern and northern extent and depth of these subglacial granitic bodies are subject to significant uncertainty (Ferraccioli et al., 2009).

The youngest rocks in the Wilkes Subglacial Basin are Ferrar Group dolerites and basalts, which intrude and overlay Beacon Supergroup siliciclastic rocks. The extent of these rock types beyond their very limited outcrops

is estimated using the previous interpretation of geophysical data in Bertram et al. (2018), with more southern limits mapped according to Jordan et al. (2013a). Jordan et al. (2013a) show a greater extent of Beacon/Ferrar Group rocks compared to the interpretations of Studinger et al. (2006) in Southern Victoria Land; we favour this more recent interpretation. Ferrar Group dolerites and basalts are isotopically well-constrained and consistent, with mean $\varepsilon_{Nd}$ values of ~-6 (e.g. Elliot et al., 1999; Hergt et al., 1989; Molzahn et al., 1996). The $\varepsilon_{Nd}$ values in

Beacon Supergroup rocks are much more varied and sparsely measured, with mean values around -12 (Fleming, 1995; Bertram, 2018; Marschalek et al., 2021). Given that (i) the relative abundance of these rocks is uncertain but Ferrar sills are common through the Wilkes Subglacial Basin (Ferraccioli et al., 2009), (ii) concentrations of Nd are approximately twice as high in Ferrar Group rocks than Beacon Supergroup rocks (Bertram, 2018), and (iii) the uppermost Beacon Supergroup rocks have a Ferrar-like isotopic composition (Elliot et al., 2017), we

assume an $\varepsilon_{Nd}$ value of -7 is reasonable for a typical mixture detritus originating from Beacon and Ferrar Group rocks.

Areas between the coast and Beacon/Ferrar Group rocks are assumed to be a mixture of Lower Palaeozoic rocks (i.e., Wilson Group, Bowers Group, Granite Harbour Intrusive and Robertson Bay Terranes), which are known to be present subglacially based on Miocene sediments recovered at Integrated Ocean Drilling Program (IODP)

Expedition 318 Site U1359 (Pandey et al., 2018). Based on the most unradiogenic $\varepsilon_{Nd}$ values measured in nearby seafloor surface sediments (~-15.5; Cook et al., 2013), this local endmember may be slightly more unradiogenic than the -14 to -12 described above; we therefore opt for an $\varepsilon_{Nd}$ value of -15 here. Sedimentary basins in the Wilkes Subglacial Basin (Frederick et al., 2016) are ignored, as these are assumed to be comprised of mixtures of relatively local bedrock and are therefore accounted for using mean $\varepsilon_{Nd}$ values for each rock type.

**A.1.3 Southern Victoria Land and Transantarctic Mountains**

The central Transantarctic Mountains provide some of the most extensive rock outcrops in Antarctica (Goodge, 2020). Neodymium isotope composition measurements are also plentiful for many rock types, making this a well-constrained region. We use the mapping of Cox et al. (2023) and isotopic data compiled and referenced in Marschalek et al. (2021) to produce the $\varepsilon_{Nd}$ value map here. In the Transantarctic Mountains, there are outcrops

of the pre-Cambrian and Beacon Supergroup/Ferrar Group rocks discussed above. Other rock types with $\varepsilon_{Nd}$ measurements include the Neoproterozoic to early Paleozoic metasedimentary Skelton, Byrd, Beardmore and Liv groups (e.g. Borg et al., 1990; Borg and DePaulo, 1994; Cox et al., 2000; Goodge et al., 2008) and



Palaeozoic Granite Harbour Intrusives (e.g. Goodge et al., 2012; Borg et al., 1990; Borg and DePaulo, 1994; Cox et al., 2000).

On the East Antarctic side of the southern Transantarctic Mountains, we extend the Ferrar Group and Beacon Supergroup into the Pensacola-Pole subglacial basin based on interpretations by Paxman et al. (2019). The Precambrian East Antarctic craton, the extent of which is defined by Goodge and Finn (2010), is prescribed an $\varepsilon_{Nd}$ value of -20 as described above. In the area between rock exposures in the southern Transantarctic Mountains and geophysical data around the South Pole, relatively little is known about the subglacial geology due to a lack of data; thus, the gap was initially simply interpolated across. However, predicted $\varepsilon_{Nd}$ values at the

ice sheet margin for this portion of the Transantarctic Mountains were consistently too radiogenic (~-5) compared to $\varepsilon_{Nd}$ values of recent sediments recovered from beneath Whillans Ice Stream (~-8 to -9) and in the Ross Sea (~-7 to -8) (Farmer et al., 2006; Marschalek et al., 2021). This may be due to a bias towards exposures of the Granite Harbour Intrusive rocks, which have a mean $\varepsilon_{Nd}$ value of -4.6 (n = 10) in this region (Borg et al.,

1990; Borg and DePaulo, 1994). These rocks are likely to be more resistant to erosion than the surrounding sedimentary rocks, such as those of the Liv Group and Beacon Supergroup, meaning they may outcrop a disproportionate amount. There may also be bias towards exposure or measurement of more radiogenic beds in the Liv Group (e.g. Wareham et al., 2001).

To combat this bias, we assigned a uniform $\varepsilon_{Nd}$ value of -9 around rock outcrops, based on the limited

measurements of Beardmore and Liv Group metasediments, which likely comprise the bedrock between the Granite Harbour Intrusive batholiths (Marschalek et al., 2021). An $\varepsilon_{Nd}$ value of -9 is also comparable to two measurements of the Beacon Supergroup in the southern Transantarctic Mountains (Marschalek et al., 2021).

To the north, the youngest rocks in this region belong to the Cenozoic McMurdo Volcanic Group in Victoria Land. These are extensively sampled and allowed to vary spatially in our map, but are generally consistent with

a mean $\varepsilon_{Nd}$ value of +5.3 ± 0.6 (e.g. Avaido et al., 2015; Martin et al., 2013; Phillips et al., 2018).

### A.1.4 Siple Coast and central West Antarctica

Ice cover is extensive and geophysical data are limited in the West Antarctic interior, making it difficult to constrain the sub-ice geology there. However, till samples from beneath ice streams and in the Ross Sea provide some guidance to the likely subglacial lithologies (Farmer et al., 2006; Licht et al., 2014). Underlying much of

this region are subglacial sedimentary basins, which range in age from the opening of the West Antarctic rift system in the mid-Cretaceous through to the Pliocene or Pleistocene (LeMasurier and Landis, 1996). Studies of sediment provenance suggest which rocks may be contributing (Farmer et al., 2006; Licht et al., 2014; Marschalek et al., 2021), including:

- A late plutonic phase of the Ross Orogeny towards the Transantarctic Mountains (i.e. Granite Harbour
Intrusives, see references above);
- Cretaceous to Triassic granitic rocks (e.g. Weaver et al., 1992; Craddock et al., 2017; Pankhurst et al., 1998);
- Palaeozoic (meta-)sedimentary rocks such as the Swanson Formation in western Marie Byrd Land and equivalents in the Whitmore Mountains (e.g. Korhonen et al., 2010; Yakymchuk et al., 2015); and





•   Palaeozoic granitic rocks such as the Ford Granodiorite in westernmost Marie Byrd Land (e.g. Weaver et al., 1992; Yakymchuk et al., 2015; Korhonen et al., 2010).

Potential mixture of these lithologies in the sedimentary bed based on adjacent exposed rocks suggests an $\varepsilon_{Nd}$ value of ~-10 (Table A1). This value agrees well with that of the most unradiogenic sediments beneath the Siple Coast ice streams (Bindschadler Ice Stream), so is thought to be representative (Farmer et al., 2006). A uniform

$\varepsilon_{Nd}$ value of -10 was therefore assigned to the entire West Antarctic interior. The magnitude and extent of contribution from Granite Harbour Intrusive rocks is difficult to constrain, but as these rocks have a mean $\varepsilon_{Nd}$ value of ~-10 (Borg et al., 1990; Borg and DePaulo, 1994; Goodge et al., 2012), they may not influence the overall composition very much, so they are neglected (Table A1).

**Table A1. Endmember assignment for central West Antarctica. The term "West Antarctic sediments" refers to our approximation for all sediments likely to underlie the WAIS since the opening of the West Antarctic rift system, including pre-glacial sediments, marine and other sediments deposited during ice-free periods, and subglacial till. It is assumed the surrounding rock types forming the source rocks for these sediment types have probably remained the same throughout the Cenozoic.**

| Rock Group | Mean [Nd] | Fraction | Mean $\varepsilon_{Nd}$ |
|---|---|---|---|
| Palaeozoic (meta)sedimentary rocks | 41 | 0.55 | -11.3 |
| Palaeozoic Granitoids | 29 | 0.225 | -6.8 |
| Mesozoic Granitoids | 12 | 0.225 | -7.0 |
| Mixture (West Antarctic sediments) | 32 | - | **-10.0** |

It is generally agreed that some Cenozoic volcanic rocks also exist beneath the WAIS. The locations of these

Cenozoic volcanic rocks were initially set based on exposed volcanoes, the subglacial volcano inventory of van Wyk de Vries et al. (2017) and the survey of Luyendyk et al. (2003). Preliminary application of the algorithm also found their presence is required to explain sufficiently radiogenic $\varepsilon_{Nd}$ values offshore, assuming surrounding rocks have values of -10 as described above. The extent of these volcanic rocks is subject to some uncertainty (Andrews and LeMasurier, 2021) and volcanic rocks comprise only a small fraction of gravel-sized

detritus in the eastern Ross Sea, but it is possible these rocks have been crushed to finer grain sizes by a combination of physical weathering and glacial comminution (Perotti et al., 2017).

Even when these potential volcanic rocks were included, $\varepsilon_{Nd}$ values were not radiogenic enough to match offshore provenance observations. This may be linked to the uniform $\varepsilon_{Nd}$ value of -10 applied for West Antarctic sediment. More radiogenic granitic rocks, potentially similar to the most radiogenic granitoids in the Ford

Ranges (Korhonen et al., 2010) or on Thurston Island (Pankhurst et al., 1993), may be widespread in some areas beneath the WAIS, as unexposed Mesozoic-aged rocks are clearly present based on detrital zircon ages (Licht et al., 2014; Marschalek et al., 2021). However, without data to constrain the extent and isotopic composition of these rocks, we favour the addition of extra Cenozoic volcanic rocks to improve the match with observations. Behrendt (2013) correlated magnetic anomalies in the WAIS interior to subglacial late Cenozoic volcanic rocks.

We assume that such rocks occur at sites with high magnetic anomalies based on the ADMAP-2 compilation (Golynski et al., 2018). Although this is a crude approach, new detailed mapping of possible volcanic material or more radiogenic granitic rocks beneath the WAIS is beyond the scope of this study. Indeed, understanding the





mixture of rocks beneath the central WAIS would be greatly beneficial for future sediment provenance studies seeking to study potential past WAIS collapse events.

A uniform $\varepsilon_{Nd}$ value of +5.3 was selected for Cenozoic volcanic rocks based on the consistent isotopic signature of such rocks throughout Marie Byrd Land and Victoria Land (Avaido et al., 2015; Martin et al., 2013; Phillips et al., 2018; Futa and LeMasurier, 1983; Hart et al., 1995). We emphasise, however, that the Kriging interpolation and 10 km spatial resolution used in our map implies that small areas of volcanic material will not result in such a high value. This was an intentional effect to partially account for the uncertainty in the presence
of these subglacial volcanic rocks.

Adjacent to the Edward VII Peninsula, where outcropping rocks are sparse, the model consistently predicted $\varepsilon_{Nd}$ values that were too radiogenic in comparison with seafloor surface sediment values which reach a mean of -11.1 ± 0.4 (n = 8) (Simões Pereira et al., 2018; Carlson et al., 2021). This was likely a result of an underrepresentation of the Swanson Formation on the Edward VII Peninsula in western Marie Byrd Land, as
this rock group shows a close geochemical affinity with the glacimarine shelf sediments in this area and may be widespread beneath the WAIS here (Simões Pereira et al., 2018). It was therefore assumed that the Swanson Formation, with an $\varepsilon_{Nd}$ value of ~-12 (Korhonen et al., 2010; Yakymchuk et al., 2015), lies beneath most of the ice on the Edward VII Peninsula.

### A.1.5 Amundsen and Bellingshausen Sea drainage sectors

Rock exposures around the Amundsen Sea embayment are particularly limited. Granites inferred to be present are assumed to be extensions of Mesozoic outcrops along this coast (Jordan et al., 2023). Cenozoic volcanics are extended, largely under Thwaites Glacier, using interpretations of magnetic, gravimetric, and topographic data (Jordan et al., 2023) as well as airborne radar data that indicate enhanced subglacial melting due to elevated geothermal heat flux (Schroeder et al., 2014). Mafic rocks are assumed to include gabbros of Cenozoic age
(Simões Pereira et al., 2020; Jordan et al., 2023), similar to Dorrell Rock near Mt. Murphy (Rocchi et al., 2006). The bedrock surrounding these intrusions is assumed to be a mixture of typical West Antarctic rocks and assigned an $\varepsilon_{Nd}$ value of -10 as described above.

A lack of rock outcrops and geophysical data translate to large uncertainties in the Bellingshausen Sea drainage sector of the WAIS (Fig. 2b). The Mesozoic granites and Antarctic Peninsular Volcanic Group present in the
Antarctic Peninsula region, including the coastal region of the eastern Bellingshausen Sea, are extended based on continuation of magnetic anomalies, but there are no outcrops to confirm this assumption is correct. Values are predominantly set to -10 as described above for the WAIS interior, assuming a similar mixture of rock types present. This matches with the $\varepsilon_{Nd}$ values of sediments from beneath the floating terminus of Pine Island Glacier, which drains part of this region into the SE Amundsen Sea embayment (Simões Pereira et al., 2020).

### A.1.6 Antarctic Peninsula

Rocks are relatively well exposed along the Antarctic Peninsula, with lithological extrapolation between outcrops possible using geophysical data (Golynski et al., 2018) and geological mapping (Burton-Johnson and Riley, 2015). The $\varepsilon_{Nd}$ values of many rock types are, however, relatively poorly constrained (see references in





Simões Pereira et al., 2018 for available measured rock $\varepsilon_{Nd}$ values). Fortunately, there is relatively little

variation in Nd isotope compositions in this geologically young region, with $\varepsilon_{Nd}$ values commonly ranging from
~-6 to -2, although some rocks have $\varepsilon_{Nd}$ values clustered around ~-10 to -8 and ~+1 to +3 (Simões Pereira et al.,
2018 and references therein). However, a relative lack of variation means uncertainty here is unlikely to
substantially influence our results.

**A.1.7 Western Weddell Sea (West Antarctic side)**

The extent of the Haag nunatak block is taken from Jordan et al. (2020). It is possible this is overlain or intruded
by rock types representing a continuation of Jurassic volcanic/magmatic activity recorded in rock outcrops on
the Antarctic Peninsula. Furthermore, provenance data suggest that old (meta)sediments may be present on top
of the Haag nunatak block as the mean $\varepsilon_{Nd}$ value of ~-7.4 for these rocks (Storey et al., 1994) is more radiogenic
than that for seabed sediments recovered offshore from this part of the Ronne Ice Shelf (Williams et al., 2017).

Most rocks around the western Weddell Sea embayment are Cambrian-Permian sedimentary rocks, Cambrian
volcanics or Mesozoic granites. Exposures are extended based on Jordan et al. (2013b), assuming outcrops are
broadly representative of rocks present beneath the ice. The Cambrian volcanics and Mesozoic granites have
been isotopically characterised, with respective mean $\varepsilon_{Nd}$ values (±2SD) of -3.5±4.3 (Curtis et al., 1999) and -
6.6±3.4 (Pankhurst et al., 1991; Borg and DePaulo, 1994; Craddock et al., 2017), but no known $\varepsilon_{Nd}$

measurements of the widespread Cambrian-Permian sedimentary strata have been performed. An $\varepsilon_{Nd}$ value of -
13 is assumed to be representative of these rocks, based on temporal equivalents in the southern Transantarctic
Mountains, such as the Byrd Group and non-volcanic beds of the Liv Group (Borg et al., 1990; Goodge et al.,
2008; Marschalek et al., 2021). Neodymium isotope compositions of offshore sediments agree with this
hypothesis (Williams et al., 2017). Beacon Supergroup and Ferrar Group outcrops in the Weddell Sea drainage

sector follow the published mapping by Paxman et al. (2019) (see Transantarctic Mountain section). The extent
of the Dufek intrusion (part of the Ferrar Group) is based on Ferris et al. (1998). Although we include the
Weddell Sea drainage sector of the WAIS in our mapping, we do not include predictions for Weddell Sea
offshore sediments in our results as these will also be influenced by supply of glacigenic debris from East
Antarctica; i.e., from sources which lie outside of our compilation area.

**A.1.8 New seafloor surface sediment data**

To evaluate whether TASP produces accurate results, we compare the model predictions with seafloor surface
sediment $\varepsilon_{Nd}$ values. To improve the spatial coverage of these measurements, we include ten new measurements
from the Ross Sea (Table A2).





**Table A2. Neodymium isotope data for new surface sediment samples in the Ross Sea.**

| Site | Sample depth | Latitude | Longitude | Water depth (m) | $^{143}Nd/^{144}Nd$ | $\varepsilon_{Nd}$ | $\pm 2\sigma$ S.E. | $\pm 2$ S.D. |
|---|---|---|---|---|---|---|---|---|
| G837 | Surface grab | -67.70 | 175.33 | 3430 | 0.512341 | -5.80 | 0.000010 | 0.29 |
| Procedural replicate | | | | | 0.512358 | -5.47 | 0.000011 | 0.27 |
| Procedural replicate | | | | | 0.512363 | -5.37 | 0.000010 | 0.27 |
| Procedural replicate | | | | | 0.512339 | -5.84 | 0.000010 | 0.29 |
| A524 | Surface grab | -73.33 | 187.20 | 3566 | 0.512336 | -5.90 | 0.000008 | 0.29 |
| A461 | Surface grab | -73.53 | 171.37 | 564 | 0.512326 | -6.08 | 0.000011 | 0.29 |
| A452 | Surface grab | -75.58 | 186.70 | 1253 | 0.512283 | -6.92 | 0.000010 | 0.29 |
| A523 | Surface grab | -73.57 | 184.22 | 2762 | 0.512252 | -7.53 | 0.000009 | 0.29 |
| E194 | Surface grab | -71.30 | 170.00 | 106 | 0.512921 | 5.51 | 0.000010 | 0.29 |
| E196 | Surface grab | -71.37 | 169.67 | 320 | 0.512864 | 4.41 | 0.000010 | 0.29 |
| E183 | Surface grab | -72.31 | 170.19 | 146 | 0.512855 | 4.23 | 0.000009 | 0.29 |
| IODP U1521 | 4-6 cm | -75.68 | -179.67 | 562 | 0.512243 | -7.70 | 0.000009 | 0.26 |
| IODP U1522 | 2-5 cm | -76.55 | -174.76 | 558 | 0.512252 | -7.53 | 0.000006 | 0.29 |

The methods used to generate these new data are identical to those described in Simões Pereira et al. (2018) and Marschalek et al. (2021). Briefly, the <63 μm fractions were leached to remove authigenic coatings, digested and subject to standard ion exchange chromatography to isolate the Nd. The $^{143}Nd/^{144}Nd$ ratio was measured on a Nu instruments HR-ICP-MS in the MAGIC laboratories at Imperial College London. Measurements of BCR-2

processed alongside samples were consistently within error of the published value (Weis et al., 2006) and blanks were 13 pg.





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
