# Peer review of "Quantitative Sub-Ice and Marine Tracing of Antarctic Sediment Provenance (TASP v1.0)"

_Geoscientific Model Development, 2024_

## Author Comment (AC1)

**Response to review by Stewart Jamieson**

Prof. Jamieson's comments are in black, our response is in blue.

This paper introduces and then tests a computational framework for predicting the provenance of sediment delivered from the Antarctic continent to the Antarctic continental shelf (and slightly beyond) on the basis of interpreting ice sheet model output. This work fills a significant gap in capability in terms of helping understand the pathways that detrital particles take as they are eroded subglacially and then transported to and beyond the ice sheet margin and theoretically allows the tracing to be completed using any ice sheet model output. The model takes into account the movement of particles such that the neodymium isotopic composition of the material is computed – this is beneficial because it can be compared directly to sediment collected offshore in a number of locations. Thus the framework should enable erosion and sediment transport and thus sediment provenance to be computed under different ice sheet regimes which should therefore produce different maps of neodymium compositions – this will allow certain ice sheet models to be ruled in or out based on their fit to measured neodymium compositions. The framework incorporates a set of appropriate transport and concentration processes although it has to make some assumptions as it does so. Processes include glacial erosion, transport subglacially, movement and rain-out from icebergs, ocean bottom currents and downslope sediment transport based on slope and the overall result is a seafloor map of Nd isotopic composition.

The code and dataset for comparison is openly available and well documented - all very clear.

We thank Prof. Jamieson for his thoughtful review and are glad that recognises that TASP fills a significant gap in capability and necessarily must make some assumptions to do so. We are also pleased that the code and dataset is recognised as well documented.

General Comments:

The paper is largely well written and easy to follow and is largely well structured with useful figures. I have some relatively minor comments on the science and writing which would be good to see addressed.

First, in places the paper mixes description of the model with descriptions of the test area where it is being applied. For example at lines 129-135 there is discussion of the area that you examine in West Antarctica. However, for a paper that is introducing a framework/piece of software that should be applicable to anywhere, I think it would be beneficial to fully describe the model framework itself before then showing how it can be applied to a particular region. I think this means reviewing carefully the introductory and methodological sections to make sure that the framework is fully introduced and then you can roll into a description of the particular ice sheet model being tested and the particular region being tested. This would make the model easier to understand and potentially easier to apply to other areas of interest.

The issue of the focus on the case-study vs broader applicability was also raised in the review by Prof. Aitken. We agree that the intermixing of the model description and our application to Antarctica resulted in a lack of clarity in the previous submission. In response, we have carefully removed any text referring to Antarctica or Nd isotopes from the Introduction and Methods sections. This now separates the model description (Section 2) from application of the model to

Nd isotope compositions in Antarctica (Sections 3 and 4). We thank the reviewer for this idea as it improves the structure of the manuscript considerably.

Second, I would like to see a clearer discussion of uncertainty or a clearer capability to embrace uncertainty within the framework. This is because at the moment particular parameter choices are made to fit the particular model area but there is no real discussion of the extent to which the result would vary if particular parameters were adjusted. For example, which parameters perturb the result most strongly? Or in other words, are there particular processes that dominate the result? It would be great to see some more discussion of that. In addition, it would be useful to see a table of parameter choices made here as this would serve to show in quick summary what was done, but would also show clearly what parameters are generally changeable within the framework so that people applying it to other areas can then make their own adjustments. Much of that information is in the user's guide but a table in the paper would give that additional help to readers.

The issue of parameter interdependence and sensitivity is also raised in Prof. Aitken's review.

We acknowledge that there are a large number of parameter choices necessary for TASP to function. We have presented an example application of TASP and shown sensitivity analysis to individual parameters (Fig. 8). There were no parameters tested that impacted RMSE by more than ~10%, so we add a sentence stating that this means we are confident that the largest influence on results is the input $\varepsilon_{Nd}$ map (new line 617-620).

Furthermore, the development of TASP over several years has necessarily involved tuning parameters multiple times. There has also been experimentation with a range of values for other parameters not formally tuned here in previous versions of TASP, which gave a 'feel' for their sensitivity. We agree that for an application of TASP an exploration of parameter inter-dependence should be undertaken and now make this recommendation to TASP users in the text (lines 619-620; for example, through a Latin hyper-cube ensemble approach) – this is something we intend to undertake in our own application studies that will follow this publication.

The existing Table 1 includes a list of parameter choices which are input into the code. We add the bottom current velocity threshold parameter and interpolation distance around different marine transport estimates to this table, which now contains every parameter used in the model. We also add a "tuning range" column, to show the range used to tune relevant parameters.

Specific Comments:

Beyond these comments, most of my suggestions are minor – they are outlined below by line number:

Equation 2 and associated text – please explain/justify why that particular erosion rule was used. Other erosion rules are available (e.g. Herman et al, erosion under an alpine glacier) – would the results differ significantly? (link to discussion about uncertainty perhaps).

We now state that: "...various erosion laws have been employed to recreate glacially eroded landscapes, typically also using an erodibility constant and the ice velocity raised to an exponent, often ~2 (e.g. Herman et al., 2015)." (line 109-111) and that "Here, we opt to use the erosion law in Equation 2 as is has been proven to reproduce reasonably accurate modern

Antarctic topography starting from a pre-glacial landscape (Pollard and DeConto, 2019), and because using a different erosion law would be unlikely to result in a significantly different pattern of erosion potential, which broadly scales with basal ice velocity and increases towards the ice margin." (lines 113-116).

Although we do not examine results with different erosion laws, there is no clear reason why a different erosion law would be preferred. We feel that all erosion laws result in the same general distribution of erosion potential (i.e., increasing towards the faster-flowing ice sheet margin), so this is unlikely to be a major source of uncertainty in our estimate offshore.

175: here you discuss a negligible change to the match with seafloor sediment Nd values in terms of how you choose k – this feels like a result or a point for discussion as opposed to something which should appear in the methods.

We agree and move this sentence to the parameter tuning section, providing more details: "For the terrestrial component, experiments were performed with spatially variable $k$ ('quarrying' coefficient) values for different ice drainage basins as in Pollard and DeConto (2019). However, this revealed a negligible impact on results. This is most likely because the map of $k$ used only varied over very large-scale basins (Pollard and DeConto, 2019). A finer resolution estimate of 'erodibility' might lead to different lithologies being disproportionately represented offshore, influencing provenance signatures." (lines 621-625).

177-180 – you discuss the Aitken and Urosevic approach and mention your approach differs, but you don't really say why/if your approach is more appropriate or provide evidence for why it might be better etc. – you could link that to my point about equation 2.

As suggested in the review by Prof. Aitken, we remove this sentence as it was not well phrased and a comparison to the approach in Aitken & Urosevic (2021) is not necessary here – neither is inherently better, as they have different objectives.

188: 'Standard Euler method' could do with more explanation. Also, explain why you think this is an appropriate method and also in terms of velocity, are you using basal velocity or ice surface velocity?

Changed this to "an Euler integration method" for clarity. Streamlines ("curves tangential to the velocity vector field") are an established mathematical way of calculating flow paths through a vector field and produce plots that align with modern ice flow velocities for Antarctica (Fig. 6). We also already state that it is the basal ice velocity here (line 127).

193-195: The computational demand is rather specific to a particular machine (which we have no real info about in terms of its specification). Therefore you could perhaps mention the types of CPUs. I would lose the point about using fewer CPUs and less memory – it seems obvious although I guess one option is to rephrase in such a way as to indicate whether there might be minimum specs that would allow the model to run (or perhaps to compare how long it would take to run on a more standard machine (assuming it would run on such a machine).

We now state that the Imperial College HCP cluster uses AMD EPYC 7742 2.25 GHz processors (line 135). We agree the CPU/memory vs runtime trade-off is obvious, so remove this statement.

205-210: I wonder whether we might benefit from a schematic diagram to illustrate how particles might move through a grid framework (e.g. showing how particles will travel subglacially and then into the marine realm). It could have some grid cells schematically drawn out with a start and end point for the particle. to show how these values are calculated as it moves through different processes. This schematic could be done just for this component of sediment transport, or another option would be to schematize the entire sediment transport process from source to iceberg to deposition in the ocean - this might help us understand the overarching structure of the processes being accounted for in TASP.

We thank Prof. Jamieson for this idea and now include a schematic diagram illustrating how debris transport is approximated in TASP (Fig. 1). We felt that limiting this to the subglacial component and including something with grid cells would not add much, especially considering the new addition of the synthetic example suggested in Prof. Aitken's review (Fig. 3). We therefore opt for cross section cartoon.

210: steady-state ice flow is assumed. Please discuss this assumption and its validity either here or in the discussion section later.

We add the following to this paragraph (lines 171-176), which describes the validity of the assumption: "The time between debris entrainment and deposition offshore may in reality be hundreds to thousands of years, depending on ice velocities. However, the primary goal of TASP is to reproduce long-term, large scale provenance patterns, such as interglacial ice sheet configurations smaller than that at present; for these, ice flow will be broadly consistent for thousands of years. At the temporal and spatial scales of interest here, the approximations used are considered sufficient to capture the broad-scale trajectories of debris under the ice sheets."

222-227: This feels a little out of place/tacked on. It could be mentioned earlier where you mention the erosion rule that is implemented, or it could be saved for discussion in the discussion section later.

We agree and move this text to the paragraph where erosion laws are discussed (lines 108-116).

233: Its not immediately clear why debris distribution in the ice column is in a section on ocean surface currents and iceberg rafting - move as appropriate or flag at the outset why the debris in the ice column is related to the iceberg processes.

We now include a sentence near the beginning of this section: "If debris is incorporated further from the edge of an iceberg, more melt needs to occur for it to be released, potentially allowing for a longer transport distance." (line 193-195)

Figure 4: Fig. 4 is a result figure, but also it doesn't really tell us whether the fit is good or not. Thus is it a helpful figure right here? Perhaps Fig. 4b is relevant here, but fig 4a does seem to be a result figure which could be saved for later when results of the tests on the specific location are presented.

We move this to our case study results section as suggested (new Fig. 9).

272: A value of 4 m debris-rich ice layer thickness. Is this also based on observations of debris layer thicknesses at all?

We tune this parameter within the range of observed thicknesses (2-12 m), and here (line 231) refer to the later section where this is discussed in detail (Section 3.2).

610: It is not indicated why/how you know this is a 'realistic-looking' distribution. Please elaborate.

This point was also raised in the review by Prof. Aitken. We re-write this paragraph (lines 462-464), now stating that results are encouraging because sedimentation decreases with distance from the sediment source, as expected.

653: TASP is better in deep waters. Better in comparison to what?

We rephrase this sentence: "TASP also performs better in deep water locations than for continental shelf sites" (lines 710-711).

666: You could quantify some of the values (predicted vs. measured) so that we understand the fit. Overall, if quantification at particular core sites is being done, then a proper description of $E_{nd}$ distribution would be good as part of the example science question you are addressing by applying TASP.

We now quote some core site predictions vs measurements for sites where down-core records exist to give an idea of the accuracy of TASP (lines 697-701): "We achieve a close match to surface sediments from specific core sites with down-core records, including International Ocean Discovery Program Site U1521 (Marschalek et al., 2021; measured = -7.7, TASP prediction = -8.0), Integrated Ocean Drilling Program Site U1361 (Cook et al., 2013; measured = -11.5, TASP prediction = -13.3) and Site PS58/254 (Simões Pereira, 2018; measured = -3.0, TASP prediction -3.7)."

As suggested, we also include a brief description of the $\varepsilon_{Nd}$ value map predicted by TASP (lines 688-692): "This produces a map of $\varepsilon_{Nd}$ values where the most radiogenic (least negative) values are found around the Antarctic Peninsula and along the Marie Byrd Land coast (Fig. 11g). Slightly less radiogenic values are found in the Bellingshausen and Ross seas. The East Antarctic coastline offshore of George V Land produces the least radiogenic values, although the influence of more radiogenic debris transported from the east is clear (Fig. 11g)."

Figure 12: Can core-top sites be shown? I don't know how much of this map is actually constrained by measurements.

We thank Prof. Jamieson for this useful suggestion and add core sites to the figure (new Fig. 14).

Section 5 (Conclusions): The conclusion could be clearer about the key processes incorporated in TASP before it gets into particular science findings. Thus add a few points about the processes at the start of conclusions to properly say what the framework does. This is important because it's a GMB paper which needs to therefore show the key points of any model before also showing any location specific results etc.

We add sentences summarising the general structure of TASP (lines 834-839). "Debris is incorporated and routed at the ice sheet bed based on ice sheet model results and an erosion law. Marine detrital particle transport mechanisms include representations of surface currents,

which are used to approximate iceberg trajectories. Bottom currents redistribute sediment if a velocity threshold is reached, and gravity flows transport material downslope. These estimates are then used to make a provenance proxy map across the seafloor, thereby directly predicting sediment core data for a given modelled ice sheet extent."

---

## Author Comment (AC2)

**Response to review by Alan Aitken**

Prof. Aitken's comments are in black, our response is in blue.

This manuscript provides a numerical approach explicitly to connect offshore sedimentary provenance records in glacial sediments with their source regions onshore, accounting for variation in sediment productivity and transport. The sediment transport problem is significant as it underpins our knowledge of cryosphere in past climate and therefore guides our ability to interpret past sea level and predict future sea level.

This problem has been approached with a range of techniques from educated guesswork to proximity studies and probabilistic assessments, and spatial modelling of individual parts of the transport system (e.g. subglacial erosion and sediment transport or ocean transport) but has not been comprehensively tackled from source to sink as is done here. As such the approach presents a unique addition to the ability to model such systems in their totality.

We thank Prof. Aitken for his detailed and useful review and are glad he recognises the uniqueness and value of TASP in trying to simulate the entirety of the system.

I have several key comments

1 - The paper is not written in the best way for the journal. I would advise a general rewrite with a stronger focus on the new approach, and less on the case-at-hand in which the authors at times get bogged down in details of the case-study and lose sight of the main goal for GMD (and also for uptake of the approach) which is to focus on the approach, its capacity and its veracity.

We extensively restructure the manuscript, removing all mention of the Antarctic Nd isotope case study from the Introduction (Section 1) and Model description (Section 2). We feel this helps prevent the manuscript getting too bogged down in case-specific detail by providing solely a description of how the model works before mentioning specifics.

2 - The degree of case-specific choices in the model is higher than I expected and I am concerned that this might limit the broader application that could make this a truly useful tool (see detailed comments). For example it is not clear if this model could, or could not be applied easily to Greenland or a model from the Pliocene. For GMD I think a more generic standpoint is needed. A simple synthetic model test case might add a lot if the authors can do so.

TASP was designed specifically to understand provenance signatures around Antarctica. Application to Greenland (or palaeo northern hemisphere ice sheets) would, in theory, be possible. However, some edits to the code would likely be required to account for the different geographic setting. Furthermore, Greenland Ice Sheets may have been mostly land-terminating during past interglacials. TASP focuses on marine processes with no representation of fluvial systems, so is not well-placed to predict the provenance signature of such an ice sheet. We now include a mention of the applicability of TASP to other ice sheets on lines 528-531.

TASP is readily applied to past Antarctic ice sheet simulations, as (alongside spatial coordinates) it only requires bed elevation, basal shear stress and basal ice velocities with horizontal directional components from the ice sheet model. There is an option ("palaeo") which, if selected, means TASP does not compare the results of the provenance tracing to

seafloor surface sediments and instead outputs just a predicted map of the provenance tracer. This is now explained in lines 499-501.

We appreciate the value that a synthetic test case would add, so include an example for the terrestrial component. We argue that constructing a synthetic model test for the marine component is less valuable given that it would be difficult to impose useful artificial ocean velocities, so opt to instead move former figure 8 (new figure 4) earlier in the manuscript, as this is useful for showing that offshore transport by the surface current method produces a sensible result with the amount of sediment from a particular sector diminishing with distance.

3 - I have some concerns about the deterministic nature of the approach and the large number of choices that are necessary for it to function. Several variables and assumptions are tested for impact and others are tuned to fit the data but the inter-relationships of parameters is not defined. In between all these moving parts there is overall a low chance that an optimal solution is found - indeed the Nd data is fitted somewhat better than a proximity-based approach but this does not indicate a minimum was found. For the paper I think a clear comment on the potential for unchecked errors to propagate through the model will suffice, but I would encourage the authors to pursue some potential ways to optimise fit to data in a more formal way.

We acknowledge that there are a large number of parameter choices necessary for TASP to function. We have presented an example application of TASP and shown sensitivity analysis to individual parameters. We add that the development of TASP over several years has necessarily involved tuning parameters multiple times. Furthermore, there has been experimentation with a range of values for other parameters not formally tuned here in previous versions of TASP, which gave a 'feel' for their sensitivity. We agree with Prof. Aitken that for an application of TASP an exploration of parameter inter-dependence should be undertaken and now make this recommendation to TASP users in the text (lines 619-620; for example through a Latin hyper-cube ensemble approach) – this is something we intend to undertake in our own application studies that will follow this publication.

4 - In my view, the true power of this model, which in each of its parts is relatively basic compared to contemporary approaches, is that it holds the whole system in one model. I would be very interested to know more in the paper about potential for modularity - for example if I wish to do detailed ocean transport but need a glacial input; or conversely if I am modelling the sediment transport in detail but need to model ocean transport to a site. TASP might be the ideal tool for this if I can "plug and play", but if it is a closed process I can't take advantage of it.

We agree that holding the whole system in one model is the true value here – as mentioned, other approaches may tackle each problem better individually, but coupling all these more accurate approaches would be impractical.

The code is structured to allow for modularity to some degree, with different functions holding the terrestrial component and each of the ocean transport methods separately. As such, it would be possible to use the output or input of another modelling approach to make use of a single aspect of TASP, provided the variables each function omitted were provided in the correct format. We feel it would be very difficult to implement any more flexibility regarding this in the code as it stands, as it would be case-specific depending on exactly which variables were wanted and an input/output.

**Detailed Comments**

**Introduction**

line 32 - in place of qualitative perhaps 'not constrained by a quantitative analysis'

Changed (line 30).

line 34-36 - I think it would be good to express the source-transport-deposition mixture problem formally. You could use Equation 1 of Aitken and Urosevic (2021) or some equivalent

We add this equation (line 38).

line 38 - I would note that there is *no clear basis* for changes in provenance to be interpreted to represent retreat and advance events unless other factors are able to be excluded (see introduction to Aitken and Urosevic (2021) and their eq 1 makes this clear). This emphasises the need for a model like TASP to define the system and reduce the potential for misinterpretation.

Although good sediment provenance studies will always consider all potential impacts on the measured signal, we add that "Unless other processes that influence provenance signatures can be eliminated, these (sediment provenance) records may be misinterpreted." (line 42-43). We agree reducing and quantifying the uncertainty of other processes is a key motivating factor for developing TASP.

line 52 - here and elsewhere 'erosion rate' should be replaced with 'erosion potential' as the true rate is never known in TASP

We thank Prof. Aitken for pointing this out and have corrected this here (line 51) and throughout the manuscript.

line 69 - I don't think the comparison to Aitken and Urosevic (2021) is particularly relevant - theirs is a probabilistic assessment of sediment production tendencies avoiding the need to model transport. There is no competition (in fact the outputs of their approach could be inputs to this approach)

We agree and remove this part of the sentence.

line 80 - An important simplification applied here is that there is no basal sediment layer. This is conceptually unappealing and also it is included in PSU ice sheet models since Pollard and DeConto, 2003 (https://doi.org/10.1016/S0031-0182(03)00394-8) and sediment transport is included in Pollard and DeConto, 2019. This layer is important as even a few metres of sediment protects the bed from erosion and spatially varying sediment cover would control strongly the provenance derived. It also can store sediments. If this truly cannot be included in TASP, then it must be made clear that the assumption is that sedimentary coverage is relatively uniform over the area.

Sedimentary armouring was carefully considered when constructing TASP. However, implementing this when using a 'snapshot' approach would be very difficult, as the thickness of the till layer will vary as a function of time, and is particularly uncertain in the past. Although the thickness of subglacial till now or in the past could be taken from the results of a time-evolving ice sheet model run (e.g. Pollard and DeConto, 2019), this would not account for the fact that till

may be incorporated into/transported by the basal ice even if active till generation is not occurring. Thus, even in an area with thick till, when no active erosion of bedrock was occurring due to armouring, till might be incorporated into basal ice and transported offshore. To truly model this, it would be necessary to step the model through time and record subglacial sediment transport so that the till provenance is known. However, this would require significant extra computational time and negate the strength of TASP as a post-processing tool that does not need to be incorporated into ice sheet model code. Knowing the composition of subglacial sediments at a resolution to make this useful would also require knowledge of subglacial geology at a far higher resolution than is currently possible.

To make this clearer, we add a paragraph discussing this problem (lines 117-125) and in some places change mentions of "generation" of debris to "incorporation" of debris, as this is essentially what is important for provenance at a given time.

line 83 - It is important to note also that subglacial fluvial transport is ignored, this too would strongly alter provenance as it can reach hundreds of kilometres into the ice sheet on short periods, and also is not necessarily aligned with ice flow.

Please see our response to former lines 213-221 below.

line 110 - Perhaps add a comment here on how it might be interfaced with complementary environment-specific transport modelling such as SUGSET or Parcels

Added that "such transport could be incorporated through interfacing with complementary modelling specifically targeting fluvial transport of subglacial sediment" at line 187-189.

**Methods**

I find the description of Nd data to be overly long for the paper, and too specific - it seems the model tracks a numerical quantity that can be safely mixed (i.e. it cannot track categorical data such as rock types, or numerical data that cannot be mixed (such as U-Pb zircon ages) ... but it could probably be used to track bulk chemistry, for example.

We feel some small description of the provenance proxy used in our case study is required, but shorten this section considerably. As Section 2 now focusses on general description of TASP, this part is now moved to the case study (Section 3, lines 509-521).

We discuss the provenance data types that could be used and the modifications to the code/input datasets required in new lines 506-508. "In the case of the categorical data (e.g. clast types) or binned distributional data (e.g. specific detrital mineral age populations), TASP would require adapting to account for multiple input maps and saving of multiple output maps, with the associated extra memory demand."

line 148 - For the purpose of this work, the choice to use offshore data to constrain onshore distribution introduces a problematic circularity...what would be the result with onshore data alone?

Throughout the development of the subglacial $\varepsilon_{Nd}$ map, circular reasoning was very carefully avoided. An $\varepsilon_{Nd}$ map based entirely on exposed geology traced subglacially using geophysical data was the starting point for this map. However, such a map is biased towards rock exposure which, in some locations, obviously contrasts with unconsolidated sediment measurements.

For instance, the interior of West Antarctica has many Cenozoic volcanic exposures, but these are known to only comprise a very small amount of sediments offshore (Andrews and LeMasurier, 2021). Similarly, exposed rock in the southern Transantarctic Mountains is dominated by relatively radiogenic granites, yet the isotopic composition of tills beneath ice streams draining this region are much less radiogenic, more closely resembling sedimentary rocks which likely surround these granites beneath fast flowing ice (see compilation in Marschalek et al., 2021).

The discrepancy between exposed and subglacial geology means that using exposed geology and geophysical data alone will not produce a map consistent with knowledge from provenance studies. To incorporate knowledge from sediments whilst making as few assumptions about unknown subglacial geology as possible, an estimated uniform Nd isotope composition was applied only in areas where there is a known discrepancy (as described above), and applied in the simplest way (i.e., as a uniform value).

We note that we were incorrect in referring to these constraints as "offshore", as the compositions were, in fact, predominantly informed by unconsolidated sediments recovered from subglacial settings along the Siple Coast (Farmer et al., 2006). These ice streams drain the southern Transantarctic Mountains and West Antarctic interior, where likely discrepancy in the isotopic composition of exposed rock and likely subglacial rock was highest. We therefore change references to "offshore" constraints to "unconsolidated sediment" constraints to reflect this important distinction. We also remove mention of "inverse" constraints in Fig. 5 (now "sedimentary" constraints), as we are actually using subglacial sediments to inform the composition, not offshore sediments.

line 152/153 - uncertainty here should probably be confidence

Changed (line 538).

line 167 - erosion potential as it is not realised as a rate

Changed throughout manuscript.

line 169 - eq 2 - sedimentary armoring of the bed is neglected. This limitation should be recognised as it is a common process to include in subglacial sediment models - Q could this be included?

Please see our response to the comment on former line 80.

line 177 - the choice of erosion scaling I think is not very important and neither is model resolution - I don't think this paragraph adds much to the paper

We agree and remove this paragraph.

Figure 3 - can we have a zoomed in view of the streamlines?

Added (inset in Fig. 6b).

line 190 - 195  A comment here (or perhaps in discussion) is needed for how TASP might scale up to a more dynamic model, or an ensemble. 8 hours is not too much to ask, but if you wanted

even to do 20 or 30 models it would become a problem. Perhaps a representative random sample of points would suffice?

To reduce the computational load, we include an option to reduce the ice sheet seed locations to only those over a certain basal ice velocity threshold, assuming that slow-flowing areas will have a smaller impact on the total debris load at the ice sheet margin (lines 129-131). However, the bulk of the computational demand arises from the surface and bottom current tracing. It is tempting to reduce the number of seed locations here, but we found doing so had a very clear impact on results. This is because the method used relies on multiple ocean streamlines crossing the same ocean cell to obtain a mean, and any reduction in ocean streamline seed locations increased the stochasticity of results in any given cell. Although this does indeed make large ensembles prohibitive, we find continual improvement using up to the maximum possible number of seed locations.

We add sentences (lines 135-137) mentioning that this option was investigated, but worsened results considerably.

line 196 to 197 To have unique streamlines for each cell-outlet pair seems excessive (perhaps I misunderstand). A more efficient approach might be to accumulate sedimentary material as it flows (e.g. using D8/Dinf algorithm and a flow accumulator)

The reviewer is correct to note that the streamline calculations are very computationally demanding and a good target for improving. However, we do not feel there is a good alternative to the current approach; the D8 and Dinf algorithms are not appropriate here as flow is not downslope. In other words, it is not possible to treat erosion potential as a DEM (with a single value in each cell), with debris passing to the 'lowest' neighbouring cell(s). If the velocity u and v components were used to route detritus to the next cell(s) based on their direction, this would also produce difficulties if a D8/Dinf type approach was used. If a D8 approach was used, this would require selecting only a single adjacent cell, neglecting important detail in flow velocities that are required to get realistic trajectories. If a Dinf-like approach was used, this would tend to produce an unrealistic dispersing pattern rather than a single path through the ice sheet/ocean. The (computationally expensive) streamline calculations therefore unfortunately represent the only feasible option.

line 198 -201 this description of mixing could be better expressed with an in line equation I think

Added an in-line equation as suggested (line 147).

line 210 - While I appreciate it is a steady state analysis - if I understand correctly you treat it as instantaneous delivery. I think there needs to be some expression here of the timeframe to transport...at 0.1 to 1 km a year you might be looking at several millennia to transport the sediment to the outlet; in somewhere like the Siple Coast, that is certainly enough time for the flow to reorganise substantially

Prof. Aitken is correct in that debris transport is effectively instantaneous in TASP. We appreciate that it might indeed take several millennia from the point of entrainment in ice for detritus to reach the grounding zone/sediment core site. However, the principal goal of TASP is to look at (broadly) equilibrium ice sheet states such as past interglacial configurations, when large-scale changes to ice sheet flow are not expected for thousands of years. Flow reorganisation in regions such as the Siple Coast will, therefore, not be represented, but such changes will likely

be mostly on scales of tens of kilometres in lateral movement along the calving front, and therefore introduce relatively small error into the results relative to the uncertainty in sub-ice geology. We briefly acknowledge that we do not account for the lag between entrainment and deposition in new lines 168-176.

line 213 to 221 - I don't think you can ignore subglacial fluvial transport even in Antarctica - high pressure channels exist and are at work evacuating sediments from far inland beneath the ice sheet. I think it is sufficient to say that TASP does not currently include this process - You could add a citation to the model codes that do tackle this such as SUGSET and GraphSSeT and if these could be integrated somehow with TASP.

As suggested to be sufficient, we state that "TASP does not account explicitly for detritus transport in subglacial hydrological networks" (line 177).

We discuss our reasoning for the omission of this transport mechanism in the paragraph at lines 177-184. An acknowledgment that "Subglacial hydrological networks will, however, evacuate some small amount of sediment beneath such an ice sheet" (i.e., without significant surface melt; line 181-182) is added. We argue that available data suggest relatively low sediment fluxes through subglacial hydrological networks at present (see review of Alley et al., 2019), and that hydrological potential tends to broadly follow ice flow trajectories at the continental scale of relevance here and given significant geological uncertainties (see Willis et al., 2016).

We acknowledge these assumptions may have a large impact for ice sheets with substantial surface melt and add a reference to papers discussing SUGSET and GraphSSeT as suggested (lines 185-189).

line 225 - Similarly here I think you needn't say it is infeasible, but it is not part of TASP and that is OK, so long as if I did want to do this in detail somewhere I can still use TASP for the rest!

As suggested, we amend this sentence to simply say that such complexities are not included in TASP.

Section 2.3 - I am less familiar with the oceans modelling sphere, but I do know there are a range of codes that can handle this in the specifics such as ROMS (Eulerian) and Parcels (Lagrangian). Similarly, to the above I think TASP has a simple approach relative to the dedicated codes and does not replace them, but gives a useful complement. Some degree of comparison is warranted.

Our description of particle tracking now includes some text mentioning that the method used for ocean particle tracking is simple compared to code designed specifically for the task (lines 273-277): "As we seek to approximate many debris transport mechanisms in a single framework, the method of ocean model particle tracking described here is relatively simple compared to code designed specifically for this task such as Parcels (Lange and Sebille, 2017) or ROMSPath (Hunter et al., 2022). Such tools are more sophisticated than required for the purposes of TASP, for instance operating in 4D and accounting for particle dispersion. TASP does not, therefore seek to replace them; they are instead a potentially useful complement."

Although not suggested by the reviewer, we also add references to iceberg models (lines 262 and 291) as we feel it was an oversight not to cite examples of these more sophisticated models.

Section 2.3.3

This section shows that with detailed observational data, we can get an acceptable representation of modern-day iceberg trajectories -- but how might this perform for, e.g. the Pliocene? Does the accuracy degrade to the point where we might as well say they travel west and not east?

This is very insightful comment as we have applied TASP using the output of unpublished Antarctic palaeo ocean modelling and found that the lower quality data do indeed impact results. The direction of the Antarctic Coastal Current (and therefore most debris transport) will remain westward under most climate scenarios as it is driven by the presence of ice on East Antarctica, so this feature is robust providing data with sufficient resolution are used.

We feel that the additional discussion here of application with palaeo ocean velocities is beyond the scope of the version of TASP presented here, as the problem is largely dependent on the resolution and quality of the ocean model data used. We are currently exploring ways to apply TASP in this way in future publications.

We now hint to the point raised in our conclusions (line 855): "... use of high-resolution modelled palaeo ocean currents would be highly beneficial."

line 327-329 - this ocean-ice harmonisation process was not very clear to me

We agree the description was hard to follow and have added some lines of pseudo-code to illustrate this (lines 580-589).

line 347 - eq 5 - the format of this equation is not very clear. It would be clearer I think to split the melt rate from the transit time d/v. Also the brackets are not necessary

The equation (now number 6) was reformatted as suggested and the brackets were removed.

line 416 to 427 - Are these processes Antarctic specific or might the processes be better represented by global data or data from data-rich margins rather than sparse local data?

Bottom current transport of sedimentary particles and their sorting along flow paths have indeed been studied in various regions of the global ocean by comparing detailed grain-size data of seafloor surface sediments with bottom-current velocities measured by moorings (e.g. McCave et al., 2017, Deep-Sea Res. I). Results showed that relationships vary slightly between different ocean regions (Fig. 3 in McCave et al., 2017) and therefore we felt it was best to focus solely on Antarctic literature as this will directly relate to the processes working in glacial (Antarctic) settings with physically-eroded sediment.

line 429 - are there not problems from the sharp cutoff? I think this could be better represented as a gradual transition.

The threshold sediment particle (re-)mobilisation by bottom currents is dependent on the composition of the sediment (grain size, mineral density, particle shape, cohesiveness etc.) and reliant on accurate modelling of bottom currents. As both of these are poorly constrained, we feel estimating some relationship between the probability of sediment mobilisation and bottom current velocities would add unnecessary complexity that would be unlikely to improve results.

line 479 - 483 - Is this the same as the D8 algorithm? and it stops when all adjacent sells are above the central cell?

Yes this is the D8 algorithm. It is now stated in the text and cited (line 425).

lines 592 to 560 - can global data or studies from data rich regions support this better?

The bottom current suspended sediment layer thickness is an extremely difficult parameter to constrain. As well as this parameter likely being significantly spatially – and in several regions, including polar margins, seasonally – variable at a fine scale, our parametrisation of this system is very simplified and the value used may not have much physical accuracy. Although literature from other regions and settings was investigated, this did not offer useful constraint, and we are wary of introducing bias towards non-polar settings by relying on studies from such data-rich regions.

**Results**

In the context of the GMD journal this section is overly focused on the case study -- which is in any case not a good basis for an accuracy test as the true result is not well known. The improved data fit is fairly equivocal due to the influence of a) parameter tuning to fit the data (which I assume was NOT done for the inverse distance) and b) I would say it is (probably) not a statistically significant outcome given the scatter in the data - although I do not have a good gauge as to expected errors in eNd data, there is a lot of horizontal scatter in Figure 9.

As described above, we now separate our model description into a separate section (2). This includes a synthetic case study to assess the accuracy of the terrestrial component (Fig. 3) and an examination of the fraction of sediment predicted to originate from each IMBIE drainage basin to assess whether the offshore component produces sensible results (Fig. 4).

We feel our data fit is notably improved compared to just interpolation offshore, as the reduction in the RMSE, from 3.70 to 3.05, is large compared to the sensitivity of the tuned parameters which never exceeds ~0.2 within the range of plausible values (see Fig. 8). The IDW parameters (distance weight e and number of neighbours ng) were tuned to an old version of TASP, but this has been updated (also Fig. 12). The results (RMSE) are shown below, with optimal parameters of 16 neighbours and a distance weight of 1. It is now stated on line 721 that the IDW parameters were tuned.

| | | e | | | |
|---|---|---|---|---|---|
| | | 1 | 2 | 3 | 4 |
| ng | 2 | 4.538 | 4.546 | 4.110 | 4.113 |
| | 4 | 4.105 | 4.107 | 4.193 | 4.224 |
| | 6 | 4.110 | 4.155 | 3.882 | 3.867 |
| | 8 | 3.895 | 3.893 | 3.749 | 3.734 |
| | 10 | 3.788 | 3.768 | 3.735 | 3.732 |
| | 12 | 3.760 | 3.745 | 3.701 | |
| | 14 | 3.715 | 3.702 | 3.718 | |
| | 16 | **3.696** | 3.701 | 3.714 | 3.716 |
| | 18 | 3.703 | 3.718 | 3.732 | 3.740 |
| | 20 | 3.697 | 3.741 | | |

Although there is a lot of horizontal scatter in Fig. 9 (new Fig. 10), the majority of the error most likely comes from uncertainty in subglacial geology. We do not, however, claim statistical significance and make the edits suggested to avoid any chance of this (see response to reviewer comment on original line 648 below).

We also now add in a sentence on line 520-521 stating typical analytical errors for εNd values (~0.2 to 0.3) for other readers not familiar with this provenance proxy.

line 610 - realistic looking and reasonable results is a weak expression

We agree and re-write these sentences, now stating that results are encouraging because sedimentation decreases with distance from the sediment source, as expected (lines 462-464).

line 631 - close agreement is a bit of a stretch given the amount of scatter in the data and R-squared of just 0.58

Changed to "agrees well" (line 692)

line 648 - MSE of 3.77, while clearly worse, is fairly close to 3.05 given there was not any tuning applied. Unless you can prove statistical significance you should delete 'considerably' and also 'much' on line 649

We cannot prove statistical significance, so remove these words as suggested. However, we do feel the algorithm is worthwhile applying, as 41% of the  sites have less than a 1 epsilon unit disagreement compared to only 18% if just using IDW. We also mention that the range in RMSE for different TASP parameter choices always remains well below 3.70 (indeed, below 3.19; lines 721-726).

line 649 - I don't think you can prove outright that the transport modelling was what caused the difference, therefore delete 'therefore'

If the seafloor surface sediment data and input map are viewed as one (as in Fig. 5a), it becomes apparent that the sediments offshore are not a good match for the rocks present immediately inland (particularly apparent areas such as in George V Land). Including westward transport of detritus in the ocean is the only feasible process that could explain this, so we feel it is extremely likely that our transport modelling will be the main factor improving the match with surface sediments and argue to retain the use of "therefore" (line 724).

**Discussion**

line 674 to 676 - The need for a high resolution observational record here works against the scope of the model for long-term examples…Add a comment here on if/how this process might be represented on long timescales to match the long-term assumptions? This is particularly true of the past

The surface current method is only being used in the 'best estimate' in very few deep-sea areas; the gravity flow method dominates beyond the shelf. The only regions where the surface current method was used in the deep ocean are far from the continent in areas not covered by sedimentary records and therefore not relevant for TASP. We therefore feel this paragraph was confusing and unnecessary, so deleted it.

line 705 to 745 - This is an overly detailed accounting for a detail of the specific application and not very relevant to the development of the model. Suggest to delete or shorten considerably

This section has been shortened as suggested (new lines 772-799). However, we feel that the outliers in Fig. 9 (new Fig. 10) near the Adare and Hallett peninsulas are useful to discuss, as they highlight that the model should not be applied to sites very near the coast where fine-scale geology is important. We also feel it is important to describe why some sites were excluded from statistics, and mention that seamounts/islands were not accounted for.

line 726 - why was 200 km chosen?

"This distance was chosen based on visual inspection of areas of obvious discrepancy with seafloor surface sediment measurements (Fig. 11)." (new line 783-784).

line 750 Figure 12 -- this figure is fairly poor and seems in part to have been clipped from a previous figure. The coastline and annotations are peculiarly chunky -- suggest to use digital coastlines from IMBIE or measures

The coastline is from the ice sheet model simulation. As the data shown are the TASP output using this model simulation, IMBIE or measures coastlines are unlikely to align and would leave gaps/overlaps, which would look messy. To improve the appearance, we remove the thick coastline outline. We also mark sediment core sites, as suggested by the other reviewer (new Fig. 14).

**Conclusion**

line 783 to 786 - the model seems to have confirmed the main features of sediment transport in the ocean…at least today

We already state here that the results are for "the modern sedimentary system", so do not feel this requires further changes.

line 795 to 800 - I am less convinced by the paleo ice sheet application - it is not clear to me how the surface ocean transport can be modelled to a comparable standard without the observations and the approach has not been demonstrated with degraded data

Please see our response to the comment on Section 2.3.3. In the conclusions, we now state that "use of high-resolution modelled palaeo ocean currents would be highly beneficial" (line 855) for applications of TASP to palaeo ice sheets. We intend to discuss palaeo applications in a later manuscript.

line 802 - It might be worthwhile to point out a potential use for predictive targeting of core sites

We add "TASP also has the potential to better target potential sediment core sites for provenance studies, as the regions with the greatest sensitivity to a provenance proxy could be identified." (lines 861-863).

line 806 - In terms of proxies, the model seems to be restricted to those that can be numerically mixed, which is probably fine for Nd, but problematic for more categorical proxies listed…

The provenance proxies listed could all be used – the only adaption needed would be to create multiple input maps and save multiple tracers. For example, one input/output could be created for each detrital mineral age bin. This would, however, increase the memory demand. We add text to our case study chapter explaining this (lines 505-508).

**Appendix**

I do not include detailed comments on the appendix for reasons of length in this commentary, and it is not very relevant to the development of the TASP model, only the application.  My recommendation would be to publish the model here and the application (including this mapping) in another journal.

We prefer to publish both the model and these data together because TASP has been primarily designed for Antarctic applications. We feel it strengthens the model description to see a real-world application in the same manuscript as it allows comparison to measured data. Parameter sensitivity analysis would also be impossible without comparison to these data.

line 880 - no data regions should just be left as no data I think

If no data regions were incorporated, this would bias output to rock exposure, which is often not representative of subglacial geology (see response to comment on initial version line 148). It is therefore preferred to make some estimate in all locations beneath the ice.

line 900 - I think to include offshore data in the definition of onshore data that is then modelled to fit offshore data introduces a problem, however small its effect

Please see our response to the comment on line 148 in the initial version.

---

## Author Response (AR2)

We are pleased that our previous revisions were positively received and are thankful to Prof. Jamieson for his additional review. Please find below our response to the minor technical corrections suggested.

Comments from the reviewer are in black and our response is in blue.

At line 620: "Do make it clear that this is a test that is being applied to ensure particles end up in a sensible place prior to conducting a full assessment of Antarctic provenance in the next section. Otherwise people may think that this is the only test being done and it would seem quite cursory until you read further."

We change this sentence to: "*To ensure that TASP transports particles to sensible areas offshore, we use an idealised provenance tracer map for the IMBIE Antarctic drainage basins (Zwally et al., 2012).*" (revised line 458).

At line 690: "I wonder if you could add some indication of what the Nd provenance has been used to interpret in Antarctica? e.g., remind the reader the range of things Nd provenance could be used to understand."

We add the sentence: "*For instance, the proxy has provided evidence for East Antarctic Ice Sheet retreat in the Pliocene and Pleistocene (Cook et al., 2013; Wilson et al., 2018) and marine-based WAIS growth in the Early Miocene (Marschalek et al., 2021).*" (revised lines 515-517).

Lines 755-764: "Is that code snippet needed? It's the only one in the paper and feels a bit out of place as a consequence. I think you explained things fine in the text in terms of dealing with the gap in ocean velocities."

We are glad our description in the text was sufficiently clear and remove the code snippet as suggested (previously after revised line 581).

Lines 765-768: "Can you add citations in terms of expected iceberg pathways. I know you refer to the figure which has citations, but having the citations in the text proper is also useful."

As suggested, we add the citations on revised line 584.

Lines 806-807: "You are saying that finer scale erodibility information would produce a disproportionate result. But is it possible that the finer scale information is correct and therefore the result might not be disproportionate, but simply it would be different (and perhaps better?). In other words, this sentence feels like its saying better knowledge of the geology would not be useful and that it would be unhelpful because it would worsen the fit between the model and the measurements. Perhaps I am interpreting it wrong so feel free to clarify."

We agree this sentence was not well phrased as it was intended to communicate that better knowledge of geology would be useful and would improve the match between the model and measurements. We now state: "*A finer resolution, lithology-based estimate of 'erodibility' might lead to different rock types being represented to greater and lesser extents offshore, improving the accuracy of modelled provenance signatures.*" (revised lines 616-618).